# OPENFWI: Large-scale Multi-structural Benchmark Datasets for Full Waveform Inversion

**Chengyuan Deng**[1,2,*]    **Shihang Feng**[1,*]    **Hanchen Wang**[1]    **Xitong Zhang**[1,3]

**Peng Jin**[1,4]    **Yinan Feng**[1]    **Qili Zeng**[1]    **Yinpeng Chen**[5]    **Youzuo Lin**[1]

[1]Los Alamos National Laboratory  [2]Rutgers University  [3]Michigan State University
[4]The Pennsylvania State University  [5]Microsoft Research
{charles.deng, shihang, hanchen.wang, xitongz, pjin, ynf, ylin}@lanl.gov
qili.zeng.cs@gmail.com, yiche@microsoft.com

## Abstract

Full waveform inversion (FWI) is widely used in geophysics to reconstruct high-resolution velocity maps from seismic data. The recent success of data-driven FWI methods results in a rapidly increasing demand for open datasets to serve the geophysics community. We present OPENFWI, a collection of large-scale multi-structural benchmark datasets, to facilitate diversified, rigorous, and reproducible research on FWI. In particular, OPENFWI consists of 12 datasets (2.1TB in total) synthesized from multiple sources. It encompasses diverse domains in geophysics (interface, fault, $CO_2$ reservoir, etc.), covers different geological subsurface structures (flat, curve, etc.), and contains various amounts of data samples (2K - 67K). It also includes a dataset for 3D FWI. Moreover, we use OPENFWI to perform benchmarking over four deep learning methods, covering both supervised and unsupervised learning regimes. Along with the benchmarks, we implement additional experiments, including physics-driven methods, complexity analysis, generalization study, uncertainty quantification, and so on, to sharpen our understanding of datasets and methods. The studies either provide valuable insights into the datasets and the performance, or uncover their current limitations. We hope OPENFWI supports prospective research on FWI and inspires future open-source efforts on AI for science. All datasets and related information (including codes) can be accessed through our website at https://openfwi-lanl.github.io/

## 1 Introduction

Understanding subsurface velocity structures is critical to a myriad of subsurface applications, such as carbon sequestration, reservoir identification, subsurface energy exploration, earthquake early warning, etc [1]. They can be reconstructed from seismic data with full waveform inversion (FWI), which is governed by partial differential equations (PDEs) and can be formulated as a non-convex optimization problem. FWI has been intensively studied in the paradigm of *physics-driven* approaches [2, 3, 4, 5, 6, 7, 8, 9, 10, 11, 12]. Negative complications of these approaches include high computation consumption, cycle-skipping, and ill-posedness issues.

With the advance in deep learning techniques, researchers have been actively exploring data-driven solutions for complicated FWI problems [13, 14, 15, 16, 17]. Recently, data-driven approaches have

---

*Equal contribution

36th Conference on Neural Information Processing Systems (NeurIPS 2022) Track on Datasets and Benchmarks.

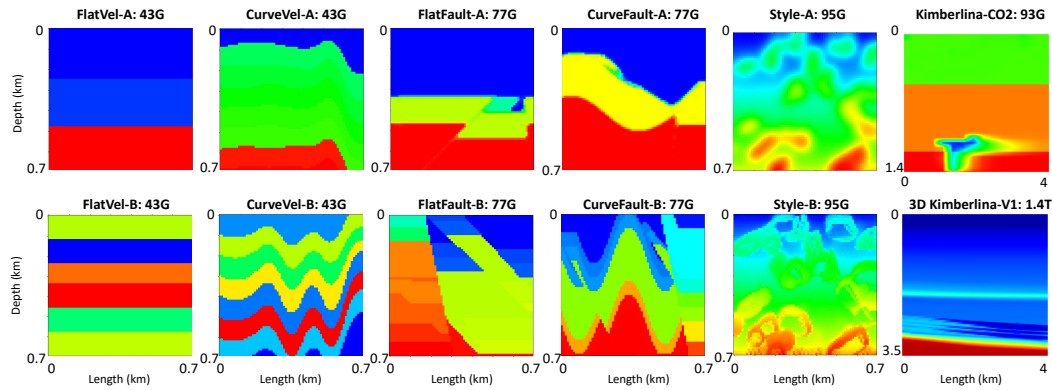

Figure 1: **Gallery of OPENFWI**, which contains one example of velocity maps from each dataset in OPENFWI.

witnessed exploration for FWI, especially on network architectures such as multilayer perceptron (MLP) [18, 19], encoder-decoder based convolutional neural networks (CNNs) [17, 20, 21, 22, 23], recurrent networks [24, 25, 26], generative adversarial networks (GANs) [27, 28, 29], etc. [30] extended data-driven FWI from 2D to 3D. UPFWI [31] leverages the governing acoustic wave equation to shift the learning paradigm from supervised to unsupervised. [32] provides a detailed survey on purely deep learning-based FWI and [33] gives a thorough overview of physics-guided data-driven FWI approaches.

Data is the oxygen for data-driven approaches, and public datasets figure prominently in developing cutting-edge machine learning algorithms. However, the FWI community currently experiences a lack of large public datasets. The existing seismic datasets [34, 35, 18, 17, 36, 37] have not been released to the public. As a result, it is difficult to perform fair comparisons among different methods.

Table 1: **Existing datasets for data-driven FWI.** The top row corresponds to our OPENFWI dataset. The symbols ✓ and ✗ indicate that the dataset has or does not have the corresponding feature, respectively.

| Dataset | Public | Multi-scale | Domains | | Geological Structures | | | | |
|---|---|---|---|---|---|---|---|---|---|
| | | | 2D | 3D | Interface | Fault | Salt body | $CO_2$ storage | Natural structure |
| OPENFWI | ✓ | ✓ | ✓ | ✓ | ✓ | ✓ | ✗ | ✓ | ✓ |
| Wang and Ma [34] | ✗ | ✗ | ✓ | ✗ | ✗ | ✗ | ✗ | ✗ | ✓ |
| Liu *et al.* [35] | ✗ | ✗ | ✓ | ✗ | ✓ | ✓ | ✓ | ✗ | ✗ |
| Araya-Polo *et al.* [18] | ✗ | ✗ | ✓ | ✗ | ✓ | ✗ | ✓ | ✗ | ✗ |
| Yang and Ma [17] | ✗ | ✗ | ✓ | ✗ | ✓ | ✗ | ✓ | ✗ | ✗ |
| Ren *et al.* [36] | ✗ | ✗ | ✗ | ✓ | ✓ | ✓ | ✓ | ✗ | ✗ |
| Geng *et al.* [37] | ✗ | ✗ | ✗ | ✓ | ✓ | ✓ | ✗ | ✗ | ✗ |

Here, we present OPENFWI, the first large-scale collection of open-access multi-structural seismic FWI datasets based on our knowledge. It contains 12 datasets, each pairs seismic data with velocity maps for different subsurface structures. Examples of velocity maps are shown in Figure 1. A comparison between OPENFWI datasets and other existing datasets for data-driven FWI is listed in Table 1. In contrast to previous datasets, our OPENFWI datasets are open-source, covering both 2D and 3D scenarios, capturing more geological structures on multiple scales. We emphasize our datasets have the following favorable characteristics:

- *Multi-scale:* OPENFWI covers multiple scales of datasets, in terms of the number of data samples and the file size. The smallest 2D dataset has 15K data samples while the largest one contains 60K samples. Four of the 2D datasets take 43GB of space each, which supports training without massive computational power. The 3D dataset occupies 1.4TB of space, therefore is usually trained in the distributed setting, further expediting the development of scalable algorithms for deep learning-based FWI.

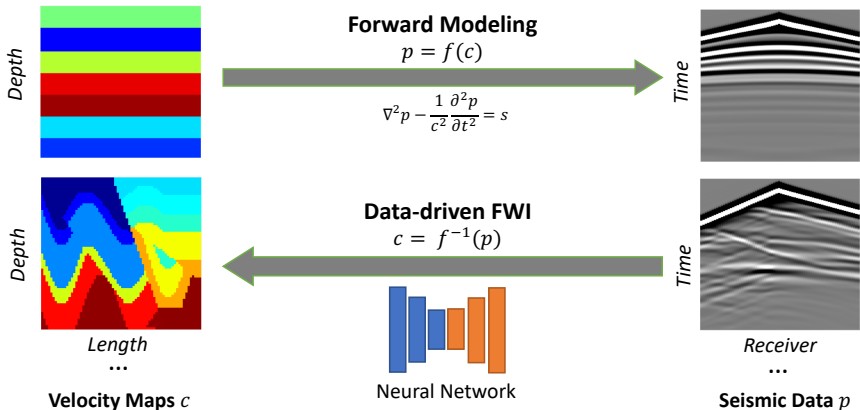

Figure 2: **Schematic illustration of data-driven FWI and forward modeling.** Neural networks are employed to infer velocity maps from seismic data while forward modeling is to calculate the seismic data using governing wave equations with velocity map provided.

- *Multi-domain:* OPENFWI empowers the research on both 2D and 3D scenarios of FWI. The datasets include velocity maps that are representative of realistic subsurface applications, such as time-lapse imaging, subsurface carbon sequestration, geologic faults detection, etc.

- *Multi-subsurface-complexity:* OPENFWI encompasses a wide range of subsurface structures from simple to complex, such as interfaces, faults, $CO_2$ storages and natural structures from natural images. The complexity is primarily measured by Shannon entropy. It supports researchers to start with moderate datasets and refine their methods for more challenging ones.

OPENFWI enables fair comparison among different methods over multiple datasets. We evaluate three representative methods (InversionNet [20], VelocityGAN [27], and UPFWI [31]) over 2D datasets, and assess InversionNet3D [30] on the 3D Kimberlina-V1 dataset. We hope these results provide a baseline for future work. For attempts on reproducibility, please refer to the resources listed in Section 1 of the supplementary materials, and the licenses therein.

OPENFWI also facilitates other related studies, such as complexity analysis, uncertainty quantification, generalization and so on. Limited by space, we briefly summarize the results of these studies and provide details in the supplementary materials. In particular, good generalizability is considered an important property of data-driven FWI, as a utopian method is expected to learn the physics rules of inversion, thus induces small errors when tested with unseen data. However, our empirical study shows existing methods suffer non-negligible degradation in terms of generalization, and it is related to the complexity of subsurface structures of the target datasets.

The rest of the paper is organized as follows: Section 2 introduces the physics background of FWI. Section 3 presents the datasets' properties concerned by domain interests. It follows in Section 4 to briefly introduce four deep learning methods for benchmarking, and demonstrate the inversion performance on each dataset. In Section 6, we initiate a discussion on the complexity of subsurface structure, the generalization performance, and uncertainty quantification, then move forward to future challenges. Finally, Section 7 concludes the paper.

## 2 Seismic FWI and Forward Modeling

Figure 2 provides a concise illustration of 2D data-driven FWI and the relationship between velocity maps and the seismic data therein. The governing equation of the acoustic wave forward modeling in an isotropic medium with a constant density is as follows:

$$\nabla^2 p - \frac{1}{c^2} \frac{\partial^2 p}{\partial t^2} = s,$$ (1)

where $\nabla^2 = \frac{\partial^2}{\partial x^2} + \frac{\partial^2}{\partial y^2} + \frac{\partial^2}{\partial z^2}$, $c$ is velocity map, $p$ is pressure field and $s$ is source term. Velocity map $c$ depends on the spatial location $(x, y, z)$ while pressure field $p$ and source term $s$ depend on the spatial location and time $(x, y, z, t)$. In this study, we focus on controlled source methods, thus the source term $s$ is given. Forward modeling of acoustic wave propagation entails calculating the pressure field $p$ by Equation 1 given velocity $c$. For simplicity, we denote the forward modeling problems expression as $p = f(c)$, where $f(\cdot)$ represents the highly nonlinear forward mapping. Data-driven FWI leverages neural networks to learn the inverse mapping as [32]: $c = f^{-1}(p)$.

## 3 OPENFWI Datasets and Domain Interests

OPENFWI datasets contain diverse subsurface structures covering multiple domains, thus supporting the study motivated by geophysics domain interests. The basic information and physical meaning of all datasets in OPENFWI is summarized in Table 2 and Table 3, including 11 2D datasets and one for 3D FWI.

The datasets are divided into four groups: "*Vel Family*", "*Fault Family*", "*Style Family*" and "*Kimberlina Family*", to address five potential topics below. The first three families cover two versions: **easy (A)** and **hard (B)**, in terms of the complexity of subsurface structures. Details on the measurement of dataset complexity can be found in Section 5.1.

Domain interests supported by OPENFWI datasets include:

- **Interfaces** that outline the subsurface structures and bound the velocity properties of rock layers [38]. To detect the interfaces, "*Vel Family*" provides velocity maps comprised of flat and curved layers that have clear interfaces. The velocity value within the layers gradually increases with depth in version A and is randomly distributed in version B.

- **Faults** caused by shifted rock layers can trap fluid hydrocarbons and form reservoirs [39]. Fault detection is crucial for identifying, characterizing, and locating the reservoirs. "*Fault Family*" includes discontinuity caused by the faults in the velocity maps, which enables the fault identification. Version B presents more discontinuities and severer velocity changes than version A.

- **Field data** from different survey areas with high diversity and complexity, which have a significant effect on the inversion accuracy [40]. "*Style Family*" enriches the diversity of the dataset by generating the velocity maps from diversified natural images, which enables the inversion of field data in general cases. Version B has the high-resolution velocity maps while those in version A are smoothed by a Gaussian filter and the corresponding seismic data contains fewer events.

- **CO$_2$ storage**, one of the most promising methods to achieve significant reductions in atmospheric CO$_2$ emissions [41] by injecting CO$_2$ into the reservoirs for long-term storage. The "*Kimberlina Family*" has two datasets simulated with high fidelity through a geologic carbon sequestration (GCS) reservoir [42]. "*Kimberlina-CO$_2$*" describes the spatial and temporal migration of the supercritical CO$_2$ plume within the reservoir, which is accompanied by timestamps within a time frame of $200$ years, and can be used for CO$_2$ storage problems, such as leakage detection and measurement.

- **3D seismic techniques** that attract increasing attention as 3D surveys have been widely implemented since [43]. The "*3D Kimberlina-V1*" dataset is the first large-scale public 3D FWI dataset. It is generated by multiple institutions [44] and supported under the US Department of Energy (DOE)-SMART Initiative [45]. It is designed and specified for the development of such techniques (not restricted to FWI). It contains a large amount of high-resolution 3D velocity maps and seismic data.

Remarkably, the velocity maps are generated from three sources: math functions, natural images, and geological reservoirs. This property enhances the diversity and generality of the velocity maps significantly. The details of the velocity map and seismic data generation pipeline are elaborated in Section 2 and Section 3 of the supplementary materials, respectively. Moreover, we provide thorough instructions on the data format, loading, and all necessary information in Section 4 of the supplementary materials.

Table 2: **Dataset summary**. Explanation of data size: Velocity maps follow (depth $\times$ width $\times$ length); seismic data represents (#source $\times$ time $\times$ #receiver in width $\times$ #receiver in length).

| Group | Dataset | Size | #Train/#Test | Seismic Data Size | Velocity Map Size |
|---|---|---|---|---|---|
| Vel Family | FlatVel-A/B | 43GB | 24K / 6K | $5 \times 1000 \times 1 \times 70$ | $70 \times 1 \times 70$ |
| | CurveVel-A/B | 43GB | 24K / 6K | $5 \times 1000 \times 1 \times 70$ | $70 \times 1 \times 70$ |
| Fault Family | FlatFault-A/B | 77GB | 48K / 6K | $5 \times 1000 \times 1 \times 70$ | $70 \times 1 \times 70$ |
| | CurveFault-A/B | 77GB | 48K / 6K | $5 \times 1000 \times 1 \times 70$ | $70 \times 1 \times 70$ |
| Style Family | Style-A/B | 95GB | 60K / 7K | $5 \times 1000 \times 1 \times 70$ | $70 \times 1 \times 70$ |
| Kimberlina Family | Kimberlina-$CO_2$ | 96GB | 15K / 4430 | $9 \times 1251 \times 1 \times 101$ | $141 \times 1 \times 401$ |
| | 3D Kimberlina-V1 | 1.4TB | 1664 / 163 | $25 \times 5001 \times 40 \times 40$ | $350 \times 400 \times 400$ |

Table 3: **Physical Meaning of OPENFWI dataset**

| Dataset | Grid Spacing | Velocity Map Spatial Size | Source Spacing | Source Line Length | Receiver Line Spacing | Receiver Line Length | Time Spacing | Recorded Time |
|---|---|---|---|---|---|---|---|---|
| "Vel, Fault and Style" Family | $10\ m$ | $0.7 \times 0.7\ km^2$ | $140\ m$ | $0.7\ km$ | $10\ m$ | $0.7\ km$ | $0.001\ s$ | $1\ s$ |
| Kimberlina-$CO_2$ | $10\ m$ | $1.4 \times 4\ km^2$ | $400\ m$ | $3.6\ km$ | $40\ m$ | $4\ km$ | $0.002\ s$ | $2.5\ s$ |
| 3D Kimberlina-V1 | $10\ m$ | $3.5 \times 4 \times 4\ km^3$ | $800\ m$ | $(4\ km, 4\ km)$ | $100\ m$ | $(4\ km, 4\ km)$ | $0.001\ s$ | $5\ s$ |

# 4  OPENFWI Benchmarks

## 4.1  Deep Learning Methods for FWI

We introduce four deep learning-based methods, InversionNet, VelocityGAN, and UPFWI for 2D FWI as well as InversionNet3D for 3D FWI, and report the inversion results as the initial benchmark. As mentioned above, UPFWI is an unsupervised learning method while the rest fall in the classical supervised learning regime. We provide a summary of each method separately as follows.

**InversionNet** [20] proposed a fully-convolutional network to model the seismic inversion process. With the encoder and the decoder, the network was trained in a supervised scheme by taking 2D (time $\times$ # of receivers) seismic data from multiple sources as the input and predicting 2D (depth $\times$ length) velocity maps as the output.

**VelocityGAN** [27] employed a GAN-based model to solve FWI. The generator is an encoder-decoder structure performing like the InversionNet, while the discriminator is a CNN designed to classify the real and fake velocity maps. It further used network-based deep transfer learning to improve the model's robustness and generalization.

**UPFWI** [31] connected the forward modeling and a CNN in a loop to achieve unsupervised learning without the ground truth velocity maps for training. The velocity maps are predicted by CNN from the seismic data and then fed into the differentiable forward modeling to reconstruct the seismic data. Eventually, the loop is closed by calculating the loss between the input seismic data and the reconstructed ones.

**InversionNet3D** [30] extended InversionNet into 3D domain. In order to reduce the memory footprint and improve computational efficiency (i.e., two of the most challenging barriers in 3D inversion), the network utilized group convolution in the encoder and employed a partially reversible architecture via invertible layers based on additive coupling [46].

## 4.2  Inversion Benchmarks

This section demonstrates the baseline results. We show the performance of three 2D deep learning methods in Table 4 and InversionNet3D for 3D FWI separately in Table 6. The network architectures of these methods and the hyper-parameters are provided in Section 5 of the supplementary materials. We consider three metrics: mean absolute error (MAE), rooted mean squared error (RMSE) and structural similarity (SSIM) [47]. MAE and RMSE both capture the numerical difference between the predicted and true velocity maps. SSIM measures the perceptual similarity between two images.

Table 4: **Quantitative results** of three benchmarking methods on 2D FWI datasets.

| Dataset | Loss | InversionNet | | | VelocityGAN | | | UPFWI | | |
|---|---|---|---|---|---|---|---|---|---|---|
| | | MAE↓ | RMSE↓ | SSIM↑ | MAE↓ | RMSE↓ | SSIM↑ | MAE↓ | RMSE↓ | SSIM↑ |
| FlatVel-A | $\ell_1$ | 0.0143 | 0.0257 | **0.9909** | 0.0118 | 0.0178 | **0.9916** | 0.0621 | 0.1233 | **0.9565** |
| | $\ell_2$ | 0.0124 | 0.0200 | 0.9901 | 0.0605 | 0.0783 | 0.9453 | | | |
| FlatVel-B | $\ell_1$ | 0.0304 | 0.0680 | **0.9356** | 0.0329 | 0.0807 | 0.9521 | 0.0677 | 0.1493 | **0.8874** |
| | $\ell_2$ | 0.0361 | 0.0751 | 0.9273 | 0.0328 | 0.0787 | **0.9556** | | | |
| CurveVel-A | $\ell_1$ | 0.0590 | 0.1231 | 0.8345 | 0.0482 | 0.1034 | 0.8624 | 0.0805 | 0.1411 | **0.8443** |
| | $\ell_2$ | 0.0574 | 0.1116 | **0.8494** | 0.0510 | 0.0976 | **0.8758** | | | |
| CurveVel-B | $\ell_1$ | 0.1448 | 0.3111 | **0.6630** | 0.1268 | 0.2618 | **0.7111** | 0.1777 | 0.3179 | **0.6614** |
| | $\ell_2$ | 0.1658 | 0.3166 | 0.6406 | 0.1428 | 0.2611 | 0.6962 | | | |
| FlatFault-A | $\ell_1$ | 0.0128 | 0.0351 | **0.9834** | 0.0868 | 0.1485 | 0.9313 | 0.0876 | 0.2060 | **0.9340** |
| | $\ell_2$ | 0.0196 | 0.0360 | 0.9830 | 0.0319 | 0.0531 | **0.9798** | | | |
| FlatFault-B | $\ell_1$ | 0.0965 | 0.1636 | **0.7323** | 0.0925 | 0.1600 | 0.7476 | 0.1416 | 0.2220 | **0.6937** |
| | $\ell_2$ | 0.1038 | 0.1637 | 0.7220 | 0.0946 | 0.1553 | **0.7552** | | | |
| CurveFault-A | $\ell_1$ | 0.0303 | 0.0766 | 0.9448 | 0.0258 | 0.0606 | 0.9613 | 0.0500 | 0.0966 | **0.9495** |
| | $\ell_2$ | 0.0331 | 0.0734 | **0.9427** | 0.0216 | 0.0505 | **0.9687** | | | |
| CurveFault-B | $\ell_1$ | 0.1705 | 0.2635 | **0.6137** | 0.1571 | 0.2427 | 0.5996 | 0.3452 | 0.5010 | **0.3941** |
| | $\ell_2$ | 0.1745 | 0.2507 | 0.6130 | 0.1583 | 0.2336 | **0.6033** | | | |
| Style-A | $\ell_1$ | 0.0625 | 0.1024 | 0.8859 | 0.0612 | 0.1000 | **0.8883** | 0.1429 | 0.2342 | **0.7846** |
| | $\ell_2$ | 0.0531 | 0.0857 | **0.9094** | 0.0645 | 0.1025 | 0.8882 | | | |
| Style-B | $\ell_1$ | 0.0669 | 0.1615 | 0.6327 | 0.0697 | 0.1108 | 0.6953 | 0.1702 | 0.2609 | **0.6102** |
| | $\ell_2$ | 0.0557 | 0.0860 | **0.7667** | 0.0649 | 0.0979 | **0.7249** | | | |
| Kimberlina-$CO_2$ | $\ell_1$ | 0.0061 | 0.0374 | **0.9872** | 0.0122 | 0.0574 | **0.9716** | \ | \ | \ |
| | $\ell_2$ | 0.0098 | 0.0400 | 0.9798 | 0.0119 | 0.0387 | 0.9527 | | | |

Table 5: **Training time** by each benchmarking method on OPENFWI datasets. Notice that the training of UPFWI and InversionNet3D occupied 32 GPUs, the rest used a single GPU.

| | Vel Family | Fault Family | Style Family | Kimberlina-$CO_2$ | 3D Kimberlina-V1 |
|---|---|---|---|---|---|
| InversionNet | 2h | 4h | 5.5h | 3.5h | 5.5h |
| VelocityGAN | 8.6h | 16h | 30h | 32h | N.A. |
| UPFWI | 30h | 60h | 60h | N.A. | N.A. |

### 4.2.1 2D FWI Benchmarks

The training parameters are identical for all 2D datasets, and the model architecture only varies a little when training using the Kimberlina-$CO_2$ dataset, noticing that its data has different input and output shapes. Two most commonly used loss functions, $\ell_1$-norm and $\ell_2$-norm, are adopted as the metrics in InversionNet and VelocityGAN while UPFWI uses a combination of $\ell_1$-norm, $\ell_2$-norm and perceptual loss as in [31]. All the experiments are implemented on NVIDIA Tesla P100 GPUs. Table 4 shows the inversion performance of three models on all 2D datasets, and Table 5 shows the estimated training time by each method on OPENFWI datasets. Note that UPFWI is not evaluated on Kimberlina-$CO_2$ because of its high computational cost. The examples of inverted velocity maps and the ground truth are demonstrated in Figure 3, where we show both successful inversion results and those unpromising. The details of training configuration and more inversion results can be found in Section 6 and 7 of the supplementary materials, respectively.

From Table 4, we observe that all three methods perform well on simple datasets such as FlatVel-A and FlatFault-A. However, there exists considerable space for improvement on difficult datasets (CurveFault-B, Style-B, etc.). Notably, VelocityGAN outperforms InversionNet on the majority of datasets by a small margin and shows comparable results on the rest. It is worth mentioning that it would take much more training time for VelocityGAN to obtain better results than InversionNet. The performance of the UPFWI velocity maps is lower than the supervised methods to a small degree because of the limited frequency band in seismic data [48]. The noticeable performance degradation for CurveFault-B indicates additional improvement on the UPFWI method would be needed.

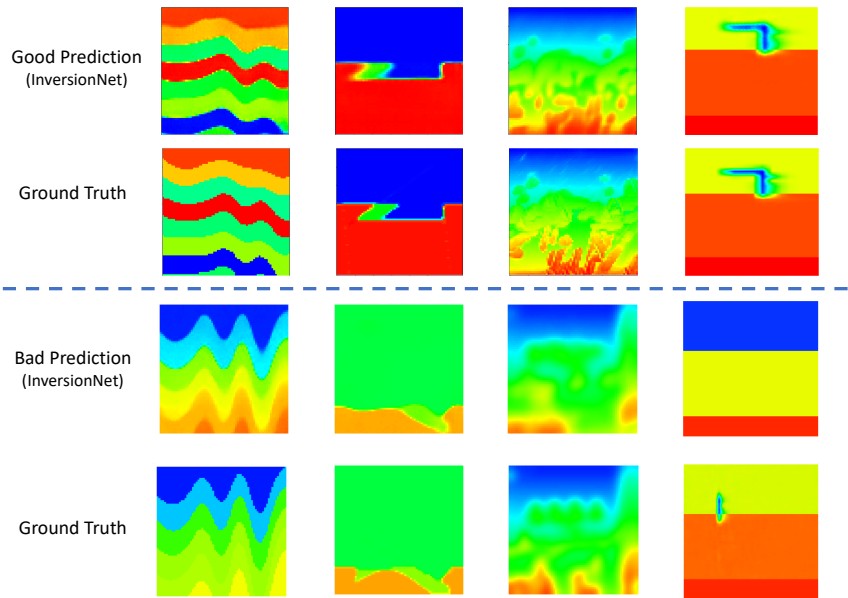

Figure 3: First two rows: Illustration of good predicted velocity maps by InversionNet and ground truth on four datasets (from left to right): CurveVel-B, FlatFault-A, Style-B, and Kimberlina-$CO_2$. Last two rows: Illustration of bad predicted velocity maps by InversionNet and ground truth on four datasets (from left to right): CurveVel-A, CurveFault-A, Style-A, and Kimberlina-$CO_2$.

### 4.2.2 3D FWI Benchmarks

Kimberlina 3D-V1 is a recently generated experimental dataset, on which only the performance of InversionNet3D [30] has been reported. In Table 6 we include the performance of InversionNet3Dx1, the shallowest version of the network, on three-channel distributions, one of which is randomly selected and the other two are symmetrical. Figure 4 explains the serial number allocation of 25 sources (channels) in the seismic data. Compared to $\ell_1$ loss, $\ell_2$ loss leads to a degradation on SSIM of 3%. More details and analysis can be found in [30].

| 0 | 1 | 2 | 3 | 4 |
|---|---|---|---|---|
| 5 | 6 | 7 | 8 | 9 |
| 10 | 11 | 12 | 13 | 14 |
| 15 | 16 | 17 | 18 | 19 |
| 20 | 21 | 22 | 23 | 24 |

Figure 4: **Spatial Placement of Sources.** Each source is the input seismic data of one channel.

Table 6: **Quantitative results** of InversionNet3D on 3D Kimberlina-V1 dataset with different channel selection strategies of seismic input.

| Training Loss | Selected Channels | MAE ↓ | RMSE ↓ | SSIM ↑ |
|---|---|---|---|---|
| $\ell_1$ | [1, 2, 14, 15, 16, 20, 23, 24] | 0.0108 | 0.0286 | 0.9838 |
| | [6, 7, 8, 11, 13, 16, 17, 18] | 0.0105 | 0.0276 | 0.9838 |
| | [0, 2, 4, 10, 14, 20, 22, 24] | 0.0107 | 0.0282 | 0.9835 |
| $\ell_2$ | [1, 2, 14, 15, 16, 20, 23, 24] | 0.0154 | 0.0306 | 0.9482 |
| | [6, 7, 8, 11, 13, 16, 17, 18] | 0.0152 | 0.0302 | 0.9476 |
| | [0, 2, 4, 10, 14, 20, 22, 24] | 0.0158 | 0.0312 | 0.9427 |

## 5 Ablation Study

In this section, we conduct intensive ablation studies including subsurface complexity analysis, generalization test, and uncertainty quantification. Each study brings insights on sharpening our understanding of OPENFWI. Moreover, We discover the current limitation of generalizability is closely related to the subsurface complexity. Limited by space, other additional experiments are described in the supplementary materials.

## 5.1 Velocity Map Complexity Analysis

Recall that the first step of data generation is to synthesize velocity maps from different priors, simulating various geological subsurface structures (interfaces, layers, faults, etc). Therefore, the velocity maps encompass different levels of complexity. We employ three standard metrics: Shannon entropy, spatial information, and gradient sparsity index to compare the relative model complexity of all 2D datasets. The spatial information captures the average boundary magnitude, and the gradient sparsity index measures the percentage of non-smooth pixels. Their math formulation is presented in Section 8 of the supplementary materials, where we also include numerical results and illustrations.

Our aim is to explore the connection between geological subsurface and performance. Therefore we demonstrate their relationship with three complexity metrics and the SSIM of three 2D benchmark methods on eight datasets in the Vel and Fault family. The reason for selecting these two families is that they follow the same generation strategy. The scatter plots and the line plots obtained from linear regression can be found in Figure 5, which indicates that the inversion performance is negatively related to the velocity map complexity, corresponding to the numerical results in Table 4. The conclusion is not surprising due to a straightforward intuition: complex velocity maps should be more difficult to be inverted from the seismic data.

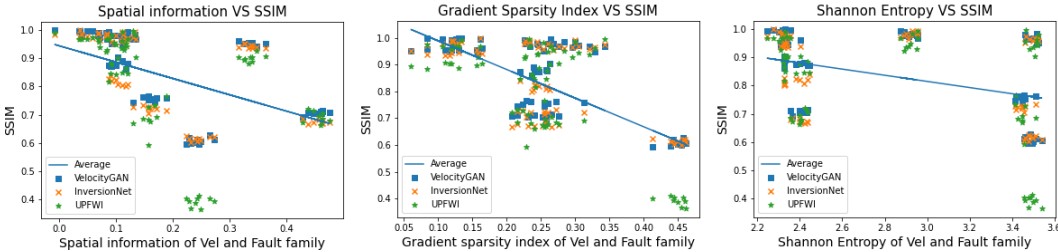

Figure 5: From left to right: three complexity metrics (spatial information, gradient sparsity index, Shannon entropy) versus SSIM. Three 2D benchmark methods (InversionNet, VelocityGAN and UPFWI) are colored in blue, orange and green, respectively. The blue line is obtained from the linear regression on the average SSIM.

## 5.2 Generalization Study

We perform pair-wise generalization tests across 10 datasets in the "*Vel*", "*Fault*" and "*Style*" families. Specifically, we select the best-trained models by VelocityGAN on each dataset ([27] claims that it shows better generalization results than InversionNet) and tested with the rest 9 datasets. The generalization performance is measured by the SSIM metric, and we obtain a $10 \times 10$ matrix illustrated in the heatmap of Figure 6, darker color indicates better generalization. We extract the relationship between these ten datasets based on the generalization performance, shown on the right of Figure 6. The results are analyzed in two-fold: *intra-domain* and *cross-domain*.

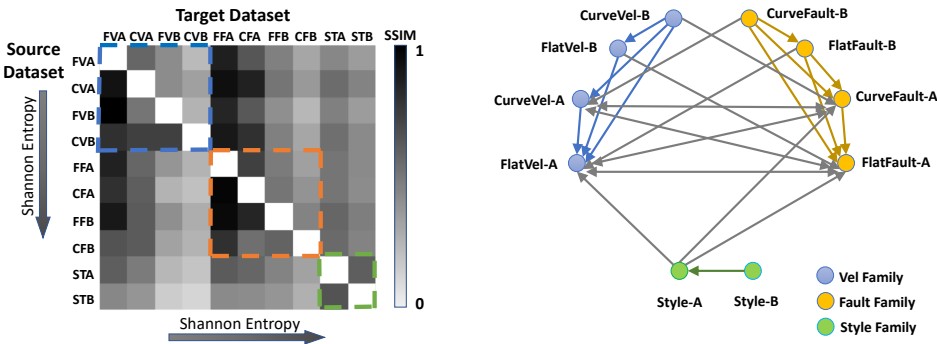

Figure 6: **Heatmap (Left) and graph (Right) of the generalization performance .** "FVA" is short for "FlatVel-A", same applies to the rest datasets. The arrow "$X \rightarrow Y$" implies the SSIM metric is above $0.6$ for model trained on $X$ and tested on $Y$.

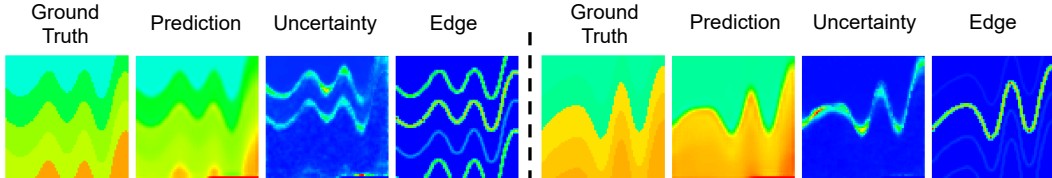

Figure 7: **Uncertainty visualization**. The uncertainty is higher on the boundaries compared with other regions.

**Intra-domain:** Focusing on the 3 diagonal blocks on the heatmap (enclosed with dashed rectangles of different colors for each data family) in Figure 6, we observe that the lower-triangle entries always have larger values than those in the upper triangle, implying that generalization from harder datasets to easier ones is more promising than the other way.

**Cross-domain:** When the source dataset is fixed, the generalization drops on the target dataset as the complexity increases. From the graph, we also observe that the degree of nodes on datasets with "A" is always higher than those with "B". The Style-B dataset has no incoming or outcoming edges to datasets in other families, thus can be regarded as a challenging dataset for generalization. More discussions on the generalization study are given in Section 9 of the supplementary materials.

### 5.3 Uncertainty Quantification

We conduct experiments on CurveVel-A to quantify uncertainty in InversionNet as a case study. As shown in Figure 7, the uncertainty on boundaries is higher than in other regions, which implies the prediction around the boundaries is more sensitive. We also observe that the uncertainty increases gradually when increasing the noise levels. Moreover, the uncertainty values of cross datasets are much higher than training and testing on the same dataset, which indicates that domain shifts lead to an increase in uncertainty. Experiment details and more results are provided in Section 10 of the supplementary materials.

### 5.4 Additional Experiments

We have conducted more experiments including the robustness test, the comparison between physics-driven methods and data-driven methods, the comparison between InversionNet and InversionNet3D and a demonstration of choosing a dataset for the target in the real scenario. All above tasks answer for major concerns in the data-driven FWI community. Limited by the space, we briefly present our findings from these experiments, more details are provided in Section 11, 12, 13, and 14 of supplementary material, respectively.

- **Robustness test**: Models are trained on 2D clean datasets but tested on noisy seismic data over multiple noise levels. Not surprisingly, degradation appears as the noise increases. We also find InversionNet is the most sensitive model.

- **Comparison between data-driven methods and physics-driven methods**: We compare two methods with respect to accuracy and computational cost. The inversion results of Data-driven methods are better by a large margin, and faster when the ratio between the number of training and test samples is less than 62.

- **Comparison between 3D simulation and 2D slices**: We train an InversionNet with 2D velocity/seismic data slices of the 3D Kimberlina-V1 dataset and compare with the Inversoin-Net3D benchmark. The results are comparable, though InversionNet3D slightly performs better (0.9652 compared to 0.9838).

- **Choosing a dataset in the real scenario**: We choose a real velocity map in [49] and generate its seismic data, then apply all twenty models trained across ten OPENFWI datasets. Only for this case, the best model trained using $\ell_1$ loss is from the FlatVel-B dataset and the best model trained using $\ell_2$ loss is from the Style-A dataset.

# 6 Discussion

## 6.1 Future Challenges

In light of the results demonstrated so far, we envisage four future challenges for data-driven FWI as listed below, where OPENFWI should be able to empower the related studies.

**Inversion for complex velocity maps:** The deteriorated performance on datasets with high subsurface complexity necessitates more advanced methods, especially those without reliance on more data.

**Generalization of data-driven methods:** The field data is usually different from the training dataset and thus good generalization is crucial for the data-driven FWI in field applications. However, the existing methods suffer non-negligible degradation on generalization. We expect more robust methods to handle data from different domains.

**Computational efficiency:** Based on our experience, UPFWI and InversionNet3D suffer from the high computational cost, which limits their potential applications. Especially for InversionNet3D, the training data is down-sampled with several channels, which may lead to the loss of information and affect its performance. More efficient algorithms are expected for these directions.

**Passive seismic imaging:** The benchmark results in this paper mainly cover the controlled/active seismic source imaging problems, but passive seismic problems is also a big sub-field. How to solve the passive imaging issues using data-driven and FWI methods requires further study and development. We conduct a preliminary test on event picking for passive data, which can be found in Section 15 of the supplementary material, to serve as a kick-off experiment for future studies.

## 6.2 Broader Impact

**Data-driven FWI:** FWI is a typical scientific problem being studied with physics-driven approaches for decades, with the rapid development of deep learning, we have seen a myriad of data-driven approaches. OPENFWI embraces this junction and brings the community with the potential of: (1) *Unified Evaluation*, (2) *Further Improvement* and (3) *Re-producibility and Integrity*, which are essential as the study on this topic evolves. We also envision OPENFWI supporting domain experts attempting to explore deep learning methods with a smooth beginning, and machine learning professionals pursuing further improvement on the current limitations.

**Future Developments:** We plan to maintain OPENFWI meticulously by releasing new datasets, and new benchmarks and serving the community with follow-up questions. There will be workshops with future updates about OPENFWI, and data competitions with more challenging data/tasks at the appropriate junction. We also appreciate any feedback from both the geophysics and machine learning communities on improving OPENFWI.

**AI for science:** Scientific machine learning (SciML) is demonstrating its great potential in various disciplines including geoscience. Compared to other fields in machine learning (such as computer vision and natural language processing), serious data challenges remain - sparse direct measurements, unbalanced data distribution, inevitable noise, etc. Our effort would hopefully shed some light on how to overcome those data challenges for SciML to enable exciting progress in typical science-rich and data-starved scientific fields.

# 7 Conclusion

In this paper, we introduced OPENFWI, an open-source platform containing twelve datasets and benchmarks on four deep learning methods. The released datasets have various scales, encompass diverse domains in geophysics, and have simulated multiple scenarios of subsurface structures. The current benchmarks showed promising results on some datasets, while the rest may need further improvement. In addition, we also include complexity analysis, generalization study, and uncertainty quantification to demonstrate the favorable properties of our datasets and benchmarks. Last, we discussed existing challenges that can be studied with these datasets and conceived the future advancement as OPENFWI evolves.

## Acknowledgement

This work was funded by the Los Alamos National Laboratory (LANL) - Laboratory Directed Research and Development program under project number 20210542MFR and by the U.S. Department of Energy (DOE) Office of Fossil Energy's Carbon Storage Research Program via the Science-Informed Machine Learning to Accelerate Real Time Decision Making for Carbon Storage (SMART-CS) Initiative. The first author would like to thank Ms. Mier Chen for valuable inputs on UI/UX design of OPENFWI.

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
