# OpenReview forum: "OpenFWI: Large-scale Multi-structural Benchmark Datasets for Full Waveform Inversion"
_NeurIPS.cc/2022/Track/Datasets_and_Benchmarks — NeurIPS 2022 Datasets and Benchmarks _

### Official Review · Reviewer_GUDA · 2022-07-27
**OpenFWI: Large-scale Multi-structural Benchmark Datasets for Full Waveform Inversion**

**Rating:** 7
**Confidence:** 3
**Correctness:** Dataset construction is sound and eva…
**Clarity:** Yes

**Strengths:**

The dataset is fairly large, easily accessible. The authors evaluate the application of the dataset to the geophysics domain and present detailed results, making the dataset and baseline implementations good benchmarks for the community to develop and evaluate against. Assuming that the code is also released in a timely manner, this would also enable good experimental reproducibility.

**Weaknesses:**

No particular weakened to point out here.

**Additional Feedback:**

n/a

**Documentation:**

Yes

**Ethics:**

No further ethical concerns

**Relation To Prior Work:**

Prior work and differences are discussed

**Summary And Contributions:**

This paper presents a large dataset to enable reproducible research in full waveform inversion (FWI) which is widely used in geophysics applications. Data is available to download, easy to access and code is expected to be released upon approval. The authors also evaluate three network architectures for FWI. Models are also evaluate for generalization and transfer learning between datasets. Detailed results are presented in the paper.

---

> ### Author Response · Authors · 2022-08-24
> **Response to reviewer GUDA**
>
> We thank the reviewer for the positive feedback. Although no specific concerns were raised during the first phase of the review, we have several improvements to update:
>
> The codes, pre-trained models, and tutorials are available. We summarize all the public resources provided to facilitate the reproducibility of *OpenFWI* as follows:
>
> - Datasets URL: https://openfwi-lanl.GitHub.io/docs/data.html#vel
> - GitHub Repository: https://GitHub.com/lanl/openfwi
> - Pre-trained models: https://tinyurl.com/bddzkxfz
> - Tutorial: https://openfwi-lanl.GitHub.io/tutorial/
> - Google Group: https://groups.google.com/g/openfwi
>
> Furthermore, to improve the quality and clarity of the manuscript, we add many additional experiments. We will update the summary of those experiments very soon.  We are delighted to provide more information if you have any concerns about the new results.

---

> ### Author Response · Authors · 2022-08-28
> **Updates on Additional Experimental Results**
>
> Based on all reviewers' comments, we have completed **additional experiments on seven topics**, including physics-driven methods, robustness test, uncertainty quantification, more 3D experiments, etc. Each one is accompanied by quantitative results and illustrations. Please refer to [Summary of Response](https://openreview.net/forum?id=7w-a8PYPlP&noteId=ZmHUoyPU-mC) (part 1-4) for the descriptions, conclusions and details of the experiments. Feel free to let us know if you have additional questions.
>
> We will upload the final version of manuscript by the end of Aug 28.

---

> ### Author Response · Authors · 2022-08-29
> **Revision Submitted**
>
> We have uploaded the revised manuscript and the supplementary materials.
>
> The manuscript is improved according to reviewers' suggestions, including re-arrangement, more discussion and illustrations, and additional experiments. We have summarized the major updates in the [Summary of response](https://openreview.net/forum?id=7w-a8PYPlP&noteId=ZmHUoyPU-mC).
>
> Please feel free with any questions.

---

### Official Review · Reviewer_HN9B · 2022-07-27
**There is a rapidly increasing demand for open geophysics datasets.**

**Rating:** 7
**Confidence:** 3

**Strengths:**

This paper provides many strong mathematical representations to illustrates the data generation process. At the same time, the authors outline five potential topics to support the study motivated by geophysics domain interest. The authors also demonstrates the baseline results and consider three different metrics.

**Weaknesses:**

For benchmark deep learning methods, I think the authors should add more details to support reproducibility.

**Additional Feedback:**

N/A

**Clarity:**

The paper is well organized and clearly written. Appropriate figures were given to make the paper understood easily. From my point of view, the work is well-done and provides an interesting topic for large-scale multi-structural benchmark datasets for full waveform inversion.

**Correctness:**

The authors provide 12 datasets synthesized from multiple sources. The data generation pipelines sounds correct. The evaluation methods and experiment design appropriate and performed correctly.

**Documentation:**

I read the full paper.  The dataset submissions include documentation and intended uses. It also include a URL for reviewer access to the dataset. For benchmarks, I think the authors should add more details to support reproducibility.

**Ethics:**

There is no ethical concerns from my point of view.

**Relation To Prior Work:**

This work is a data-driven FWI methods which differs from previous physics-driven approaches.

**Summary And Contributions:**

This paper proposed benchmark datasets for multi-structural seismic full waveform inversion. OpenFWI contains 12 datasets and has following favorable characteristics: (1) Multi-scales (2) Multi-domains (3) Multi-subsurface-complexities. The authors evaluated three representative methods (InversionNet, VelocityGAN, UPFWI) over these datasets. All three methods perform well on some datasets, while the rest may need further improvement. At last, the authors raise three challenges that can be studied with and consider the future advancement as OpenFWI evolves.

---

> ### Author Response · Authors · 2022-08-24
> **Response to reviewer HN9B**
>
> **Q1: For benchmark deep learning methods, I think the authors should add more details to support reproducibility.**
>
> We thank the reviewer for raising the concern about the reproducibility. Most importantly, we have released the codes on GitHub. Moreover, we provide a detailed tutorial for users to get started with OpenFWI easily.
>
> Here is a list of all resources to facilitate the reproducibility of *OpenFWI*, we are open to providing further details based on the feedback from our users.
>
> - Datasets URL: https://openfwi-lanl.GitHub.io/docs/data.html#vel
> - GitHub Repository: https://GitHub.com/lanl/openfwi
> - Pre-trained models: https://tinyurl.com/bddzkxfz
> - Tutorial: https://openfwi-lanl.GitHub.io/tutorial/
> - Google Group: https://groups.google.com/g/openfwi
>
> In addition, to improve the quality and clarity of the manuscript, we add many additional experiments and have interesting discoveries. We will update the summary of those experiments very soon. We are delighted to provide more information if you have any concerns about the new results.

---

> ### Author Response · Authors · 2022-08-28
> **Updates on Additional Experimental Results**
>
> Based on all reviewers' comments, we have completed **additional experiments on seven topics**, including physics-driven methods, robustness test, uncertainty quantification, more 3D experiments, etc. Each one is accompanied by quantitative results and illustrations. Please refer to [Summary of Response](https://openreview.net/forum?id=7w-a8PYPlP&noteId=ZmHUoyPU-mC) (part 1-4) for the descriptions, conclusions and details of the experiments. Feel free to let us know if you have additional questions.
>
> We will upload the final version of manuscript by the end of Aug 28.

---

> ### Author Response · Authors · 2022-08-29
> **Revision Submitted**
>
> We have uploaded the revised manuscript and the supplementary materials.
>
> The manuscript is improved according to reviewers' suggestions, including re-arrangement, more discussion and illustrations, and additional experiments. We have summarized the major updates in the [Summary of response](https://openreview.net/forum?id=7w-a8PYPlP&noteId=ZmHUoyPU-mC).
>
> Please feel free with any questions.

---

### Official Review · Reviewer_XZjm · 2022-07-27
**Review of OpenFWI: Large-scale Multi-structural Benchmark Datasets for Full Waveform Inversion**

**Rating:** 9
**Confidence:** 4

**Strengths:**

The published dataset, comprising various synthetic subsurface velocity models and generated seismic data, provides a highly valuable open source contribution for researchers and data scientists in seismology and machine learning. Not only does it allow for comparability newly developed methods to baselines, but it also allows for testing of generalization of the methods across distinct geological domain characteristics.

What is highly appreciated is the reporting of negative results as the study demonstrates limitations of the generalization of the inversion methods when increasing complexity in cross-domain experiments.

Regarding the choice of baseline methods, it is highly appreciated that an unsupervised DL-method was included. This has big impact on the applicability of the methods in real world settings.

The generation of the velocity maps documented in detail. The supplementary material is very comprehensive and helpful to get further information if needed (e.g. on the seismic datasets, on the benchmarks, and on intermediate inversion results).

Besides the generalization study, the inclusion of a complexity analysis of the velocity maps is an additional detail and makes the study very complete.


**Weaknesses:**

Unfortunately, approval for releasing the codes is still pending by the time of this review. This limits the evaluation for accessibility and reproducibility, which is an integral part for this track. Given the user-friendly website, I hope that this can be resolved during the revision period. For the time being, it does not permit a higher rating.

It is mentioned that physics-based FWI suffers from high computation costs. This is not put into relation with the computational costs of the presented DL-based FWI methods (training time of the latter are provided in the Supplementary Materials). Could the authors add a rough comparison?

The study doesn’t discuss robustness against seismic noise of the methods, which has significant impact in real world problems. I acknowledge that the focus of the work is to provide benchmark datasets for FWI, and that field data is to be added in the future, but drawing limitations regarding seismic noise and discussion of uncertainty quantification would increase the impact of the work.


**Additional Feedback:**

The channels (i.e. directionality) of the seismic stations are not well introduced. It can only be inferred from the figures in the supplementary material that 5 channels are being recorded. How are they used in the inversion methods?

It would be very insightful to see how the DL-methods perform compared to conventional physics-based method, which would be great to have included in the benchmark. However, this might be out of scope of the present work.

Will the published code include the data generation algorithms? (Both for velocity model construction as well as for seismic forward modeling.) This could increase impact of the work as it adds flexibility and researchers could generate their own models (e.g. with a source at depth, varying source frequencies, etc.).

Table 1: Add unit of time dimension, and inter-station distance and domain size

Line 180: Should be “l2-norm”






**Clarity:**

Overall, the paper is written very clearly. The Additional Feedback section below provides some individual remarks.

**Correctness:**

Both dataset and benchmark pipelines are constructed and evaluated appropriately. Results are reported correctly, including the reporting of negative results (i.e. in the generalization test).

**Documentation:**

An explicit license statement is missing for both data and code (the authors state that the code is yet to be released and discussions on the licensing are ongoing).

The data is shared through a Dropbox link. This way, maintenance is questionable. For the final submission, a data repository (e.g. zenodo) with a persistent dereferenceable identifier (DOI) would be preferred.

The creation of the velocity models is very well documented in the manuscript, including the used mathematical formulations.

Regarding the seismic data generation, the seismic forward modeling is not documented in detail (E.g., were open boundary conditions applied? What is the dimension of the elements in the mesh for FEM and the numerical step size to ensure convergence?). This can be accepted as being a standard prior art - a reference to the numerical modeling method was included.

A figure illustrating the placement of the seismic sources and receivers would be helpful to better understand the experimental design for the seismic data generation.


**Ethics:**

No ethical concerns are observed.

**Relation To Prior Work:**

The authors mention existing datasets (which they claim are not publicly released).
Prior art on FWI is well covered, spanning physics-based, data-driven, and hybrid approaches.
The used methods are appropriately referenced.


**Summary And Contributions:**

The submission provides benchmark datasets of various subsurface velocity models for inverse problems in seismology. The datasets comprise 12 synthesized velocity models, representing a variety of different styles, complexities and dimensions, and corresponding seismic data, generated from source and receiver surveys placed on the surface of the domains. The problem of Full Waveform Inversion (FWI) is introduced, and ML inversion benchmarks are generated from 4 different Deep Learning methods, including one unsupervised neural network. The performance of the benchmarks is evaluated with three different metrics, capturing numerical as well as structural differences. The submission is accompanied by a user-friendly website (https://openfwi-lanl.github.io/index.html), while the code is not yet published (license approvals still in progress).

---

> ### Author Response · Authors · 2022-08-26
> **Response to reviewer XZjm [Part 3]**
>
> **Q4: An explicit license statement is missing for both data and code.**
>
> The codes are released on GitHub under the OSS license and BSD-3 license, as required by the Los Alamos National Laboratory (LANL) and the U.S. Department of Energy (DOE), U.S.A. We have included the statement in our GitHub repository. For the data license, we choose the Creative Commons Attribution-NonCommercial-ShareAlike 4.0 International License. The license is added in OpenFWI Dataset URL: https://openfwi-lanl.GitHub.io/docs/data.html#vel
>
> **Q5: The data is shared through a Dropbox link. This way, maintenance is questionable. For the final submission, a data repository (e.g. zenodo) with a persistent dereferenceable identifier (DOI) would be preferred.”**
>
> Thanks for the suggestion. Now the data is maintained on Google Drive, and we are working with DOE/LANL on the data repository.
>
> **Q6: The seismic forward modeling is not documented in detail (E.g., were open boundary conditions applied? What is the dimension of the elements in the mesh for FEM and the numerical step size to ensure convergence?).**
>
> We have added the details of the forward modeling as follows:
>
> We follow the forward modeling algorithm described in https://csim.kaust.edu.sa/files/SeismicInversion/Chapter.FD/lab.FD2.8/lab.html. The seismic data is simulated using finite difference methods with the absorbing boundary condition and the Ricker wavelet is the source function. The original code in the link is written in MATLAB. To increase its computational efficiency and its compatibility with the neural network, we change its scheme to 2-4 (2nd-order accuracy in time and 4th-order in space) and rewrite it in Python. Sources and receivers are evenly distributed on the surface. The details of the forward modeling configuration are listed in the table (see link: https://tinyurl.com/4xvejn2f), including the grid spacing of the velocity maps, the central frequency of the source wavelet, the grid number of the boundary, and others.
>
> **Q7: A figure illustrating the placement of the seismic sources and receivers would be helpful to better understand the experimental design for the seismic data generation.**
>
> Source locations are annotated as red stars in both 2D (https://tinyurl.com/pvf5tu7p) and 3D (https://tinyurl.com/4ue3y4pw) cases. The receivers are distributed all over the surfaces and we add the description of the receivers in the captions of the figures.
>
> **Q8: The channels (i.e. directionality) of the seismic stations are not well introduced. It can only be inferred from the figures in the supplementary material that 5 channels are being recorded. How are they used in the inversion methods?**
>
> In data-driven methods, the channels are stacked together into a tensor (5,1000,70) in our data and fed into the neural network. In the first convolution layer, the 2D convolution embeds the input seismic data into a feature map of shape (32,500,70). Please refer to a diagram of this process via this link: https://tinyurl.com/2p95m5y2
>
> **Q9: Will the published code include the data generation algorithms? (Both for velocity model construction as well as for seismic forward modeling.)**
>
> For the seismic forward modeling, we use the algorithm described in https://csim.kaust.edu.sa/files/SeismicInversion/Chapter.FD/lab.FD2.8/lab.html. Below is an example of the acquisition geometry setting in the MATLAB script for the 1st source in the ``Vel, Fault and Style'' dataset. This example is added to the paper. The readers can adjust these parameters easily to generate their own data.
>
> ```
> nz=70; nx=70;
> dx=10; nbc=120; nt=1001; dt=0.001;
> freq=15; s=ricker(freq,dt); isFS=false;
> coord.sx = 1*dx; coord.sz = 1*dx;
> coord.gx=(1:nx)*dx; coord.gz=ones(size(coord.gx))*dx;
>  ```
>
> For the velocity construction of the Style Family, we use the GitHub package in https://GitHub.com/kewellcjj/pytorch-multiple-style-transfer. The Marmousi map (https://wiki.seg.org/wiki/AGL_Elastic_Marmousi) is used as the style image and the COCO dataset (https://cocodataset.org/) is used as the content images. The GitHub information is added to the paper. Unfortunately, other velocity model construction cannot be released due to the constraint of the U.S. DOE and LANL. The Vel Family and Fault Family can be constructed following the equations in the paper.
>
> **Q10: Table 1: Add unit of time dimension, and inter-station distance and domain size**
>
> The physical information of the dataset is added in the main body as Table 3.
>
> **Q11: Line 180: Should be “l2-norm”**
>
> Thank you for pointing it out! The typo is corrected in the revised manuscript.

---

> ### Author Response · Authors · 2022-08-26
> **Response to reviewer XZjm [Part 2]**
>
> **Q3.1: The study doesn’t discuss robustness against seismic noise of the methods, which has significant impact in real world problems.**
>
> - **Experiment 1: Adding noise on 2D test datasets.** Models are trained on 2D clean datasets but tested on noisy seismic data over multiple noise levels.
>   - **Conclusion:** All models are robust when noise level is low. When the noise level increases, more degradation is observed in all models. Among them, InversionNet is the most sensitive to noise.
>   - **Details and illustrations:** We add four levels of Gaussian noise with mean $\mu=0$ and different standard deviation $\sigma_{test}$ ranging from $1\times10^{-5}$ to $5\times10^{-4}$ to normalized seismic data in test set. The performance of each model is summarized in the table at https://tinyurl.com/y82cp7y6  and the figure at https://tinyurl.com/yckpauk5.
>
> - **Experiment 2: Adding noise to the 3D test dataset.** Models are trained on 3D clean datasets (without noise) but tested on noisy seismic data over multiple noise levels.
>   - **Conclusion:** Both the $\ell_1$- and $\ell_2$-trained InversionNet3D models are relatively robust to Gaussian noise with negligible degradation.
>   - **Details and illustrations:** We add four levels of Gaussian noise with mean $\mu=0$ and different standard deviation $\sigma_{test}$ ranging from $1\times10^{-5}$ to $5\times10^{-4}$ to normalized seismic data in test set. The performance of each model is also shown in the table at https://tinyurl.com/y82cp7y6.
>
> - **Experiment 3:  Adding noise on both training and test 2D dataset.** Due to the limited rebuttal duration, we only train our models on CurveVel-A with noisy seismic data to verify if it can improve model robustness.
>   - **Conclusion:** Introducing noise into training effectively improves model robustness for all three models when testing noise level is high. When testing noise level is lower than training noise level, results are mixed. Specifically, InversionNet trained with noisy data achieves better performance than the one trained on clean data, while the other two models (VelocityGAN and UPFWI) have a slight performance drop.
>   - **Details and illustrations:** We train our models on CurveVel-A with two levels of Gaussian noise ($\sigma_{train}=1\times10^{-4}$ and $\sigma_{train}=1\times10^{-4}$). The standard deviation of the noise added to the test set $\sigma_{test}$ranges from $1\times10^{-5}$ to $5\times10^{-4}$. Quantitative results are shown in the table at https://tinyurl.com/2p97ayvn. Visualization of results is provided at https://tinyurl.com/25rw68yp.
>
> **Q3.2: Drawing limitations regarding seismic noise and discussion of uncertainty quantification would increase the impact of the work.**
>
> - **Experiment 1: Quantifying uncertainty.** Due to the limited rebuttal duration, we follow [1] and conduct experiments on CurveVel-A to quantify uncertainty of InversionNet as a case study.
>   - **Conclusion:** The uncertainty on boundaries is higher than in other regions, which implies the prediction sensitivity around the boundaries. To quantify the correlation between the uncertainty and boundaries, we calculate the Pearson correlation coefficient between the uncertainty value and the gradient magnitude on edge. The value is 0.5462, indicating a moderate positive correlation.
>   - **Details and illustrations:** Following [1], we modify the network architecture by adding an additional dropout layer with a dropout rate of $p=0.2$ after each convolutional layer except the last one. The visualization of uncertainty is provided in figure at https://tinyurl.com/ycx4bjbf. The relationship between uncertainty and the gradient magnitude on edge is shown in the scatter plot (see link: https://tinyurl.com/5459u2v).
>
> [1] Kendall, A., & Gal, Y. (2017). What uncertainties do we need in bayesian deep learning for computer vision? Advances in neural information processing systems, 30.
>
> - **Experiment 2: Quantifying uncertainty on noisy seismic data.** We compare uncertainty on noisy seismic data over multiple noise levels.
>   - **Conclusion:** The uncertainty increases gradually when increasing noise level.
>   - **Details and illustrations:** We add four levels of Gaussian noise with mean $\mu=0$ and different standard deviation $\sigma_{test}$ ranging from $1\times10^{-5}$ to $5\times10^{-4}$ to normalized seismic data in test set. Quantitative results are shown in the table at https://tinyurl.com/5ysb8enj.
>
> - **Experiment 3: Quantifying uncertainty for generalization test (cross datasets).** We train InversionNet on CurveVel-A and test it on all the other 2D datasets, excluding Kimberlina-CO$_2$ due to its different data dimensions.
>   - **Conclusion:** The uncertainty values of cross datasets are much higher than training and testing on the same dataset, which indicates domain shifts lead to an increase in uncertainty.
>   - **Details and illustrations:** Quantitative results are shown in the table at https://tinyurl.com/yc76dzfj.

---

> ### Author Response · Authors · 2022-08-26
> **Response to reviewer XZjm [Part 1]**
>
> We thank the reviewer for the feedback and for raising the concerns.
>
> **Q1: Unfortunately, approval for releasing the codes is still pending by the time of this review. This limits the evaluation for accessibility and reproducibility, which is an integral part for this track.**
>
> The codes have been approved by DOE/LANL for release and are now publicly available. Together with other links, we summarize all the public resources to facilitate the reproducibility of *OpenFWI* as follows:
>
> - Datasets URL: https://openfwi-lanl.GitHub.io/docs/data.html#vel
> - GitHub Repository: https://GitHub.com/lanl/openfwi
> - Pre-trained models: https://tinyurl.com/bddzkxfz
> - Tutorial: https://openfwi-lanl.GitHub.io/tutorial/
> - Google Group: https://groups.google.com/g/openfwi
>
>
> **Q2: It is mentioned that physics-based FWI suffers from high computation costs. This is not put into relation with the computational costs of the presented DL-based FWI methods (training time of the latter are provided in the Supplementary Materials). Could the authors add a rough comparison?**
>
> - **Experiment 1 (Prediction accuracy):** We compare data-driven methods and physics-driven methods in terms of *prediction accuracy*
>
>   - **Conclusion:** Data-driven methods have better performance than physics-driven methods. For example, the SSIM of data-driven methods on the CurveVel-A dataset is 24% better than the result by physics-driven methods (SSIM 0.8442 vs. 0.6816).
>
>   - **Details and illustrations:** We perform the physics-driven methods on 500 samples from each dataset. The comparison between quantitative results of data-driven methods and physics-driven methods is given at https://tinyurl.com/4ph3by6h. The illustrations of the data-driven and physics-driven inversion results in each family are provided: [Vel Family](https://tinyurl.com/pjue2z9z), [Fault Family](https://tinyurl.com/294sps7e), [Style Family](https://tinyurl.com/3mubc7hh) and [3D Kimberlina-V1](https://tinyurl.com/2p8wxavz), respectively.
>
> - **Experiment 2 (Computational cost):** Physic-driven methods and data-driven methods measure computational cost differently. Physics-driven methods don’t need training data but are directly optimized on the test data for multiple epochs to generate the inversion results, while data-driven methods are trained on training data for multiple epochs and then applied to the test data in one trial. Thus, the computational comparison varies over different ratios between the number of training and test samples.
>
>   - **Conclusion:** All three data-driven methods are faster when the ratio between the number of training and test samples is less than 62. Typically, the ratios for all twelve OpenFWI datasets are significantly lower (about 10). Therefore, data-driven methods are 6 times faster than physics-driven methods. Note: data-driven would benefit from a large ratio between the number of training and test samples.
>
>   - **Details and illustrations:** The pre-sample computational time of physics-driven methods and the average training time of data-driven methods for each sample are given at https://tinyurl.com/yksamx2u.  We discuss the relationship between the total data-driven/physics-driven computation time per test sample and train/test ratio for different datasets in https://tinyurl.com/y84cbe7t.

---

> ### Author Response · Authors · 2022-08-28
> **Updates on Additional Experimental Results**
>
> Based on all reviewers' comments, we have completed **additional experiments on seven topics**, including physics-driven methods, robustness test, uncertainty quantification, more 3D experiments, etc. Each one is accompanied by quantitative results and illustrations. Please refer to [Summary of Response](https://openreview.net/forum?id=7w-a8PYPlP&noteId=ZmHUoyPU-mC) (part 1-4) for the descriptions, conclusions and details of the experiments. Feel free to let us know if you have additional questions.
>
> We will upload the final version of manuscript by the end of Aug 28.

---

> ### Author Response · Authors · 2022-08-29
> **Revision Submitted**
>
> We have uploaded the revised manuscript and the supplementary materials.
>
> The manuscript is improved according to reviewers' suggestions, including re-arrangement, more discussion and illustrations, and additional experiments. We have summarized the major updates in the [Summary of response](https://openreview.net/forum?id=7w-a8PYPlP&noteId=ZmHUoyPU-mC).
>
> Please feel free with any questions.

---

### Official Review · Reviewer_7wdh · 2022-07-27
**If the code was public, and the paper heavily edited, these would be useful tools for a relevant real-world problem.**

**Rating:** 7
**Confidence:** 3

**Strengths:**

The simulated datasets and (promised) inversion tools, will be useful for the development and testing on novel inversion algorithms tailored for an applied problem with much active interest.


**Weaknesses:**

I see two major weaknesses.  Firstly, the link in the paper for the code is broken. This should be fixed promptly.  If it is not, I would suggest the area chairs consider rejecting. I would downgrade my rating form a 7 to a 5 (marginal reject).

 Secondly, I found the paper very difficult to read. I will say more in the "clarity" section.


**Additional Feedback:**

N/A

**Clarity:**

I found the paper very difficult to read.  I believe the authors have done good work.  It is relevant, (to the best of my knowledge) novel, and timely. But it is not particularly complicated, and the tools are quite standard.  Thus the paper can, and should be, very simply written.

I suggest the following:

- The figures (at least the first appearance of each kind) need axes-labels.  Someone familiar with geophysical inversion will know that the velocity maps are depth-versus-distance and the seismic data are sensor-versus-time.  A Machine Learning enthusiast who encounters this paper will not know this and the authors do not tell them.

- From Equation 1 to Equation 2, $p(r,t)$ has become simply $p$.  A simple acknowledgement that the dependence on location and time are retained but simply not shown will aid a reader. Or, given that $p$ does not appear frequently in the text, just leave the explicit dependence. Also, Equation 2, has dropped the dependence on the source properties $s(r,t)$.  While this is common in geophysical inference (where, the source is often a controlled explosion or someting similar), this should be stated explicitly. In the future work section, the authors might acknowledge that there is a subfield of "passive geophysical inference" that makes use of more complex, and potentially unknown, sources. The tools herein might be applicable to such problems.

- Section 3 begins by naming 4 data groups, and then proceeds over a full paragraph to give details about the groups, many of which are repeated in Table 1.  At no point, in section 3.1, is it ever stated how the groups differ and for what each might be used.  This doesn't come until Section 3.3.  I would strongly recommend starting with the current section 3.3 use-cases and then putting a column of "domain interests" or "applications" in table 1.  That way when the groups are introduced, a reader can easily connect each group to its purpose.

- I would recommend moving section 3.2, on how the velocity maps were synthesized to the supplemental materials. Except for the short mention of the Kimberlina family/Geological resevoir, this section is technical and not relevant to most readers, who will likely be more interested in the inference tools.

- Fig. 4 is impressive, but  readers would benefit from seeing what more dramatic failures look like. Such as are shown in Figs S6-S9 in the supplemental.

- It is a lot of work for a reader to connect Table 4 and Table 3 and confirm the (unsurprising) claim: "complex velocity maps should be difficult to invert from the seismic data".  I would suggest three scatter plots of "complexity metric" vs. SSIM, with the different inverse models shown in different colors.

- The generalization study has quantified things we expect (algorithms perform best on the data sets upon which they were trained; and training on complex and testing on simple is better than vice versa).  But it has not addressed the much more interesting question of real-world relevance.  Which training set should one use to train an inverse model to work on real-world data?  Would our decisions change if we had some prior knowledge on the likely presence/absence of fault-lines, or a crude estimate of sub-surface "complexity"?  If we used multiple inverse models trained on different data synthetic sets and found different sound-speed maps how might we decide which to trust?

- In Sec 5.3 on future challenges the authors mention computational efficiency.  Up until now this had not been mentioned much in the paper.  If possible adding a metric of computational cost to Table 2 would help a reader recognize and understand the trade-off between performance and cost.


**Correctness:**

I have no reason to think the work is incorrect.  The results are plausible, as far as I know.

**Documentation:**

The broken github link for the code is problematic.  Also the "Resources -> tutorial" link seems broken on the webpage.

Otherwise, the shared data and benchmarks seem tolerable.


**Ethics:**

I have no concerns.

**Relation To Prior Work:**

The authors discussion of previous work, and their relation thereto, seems reasonable to me. But I admit I am not up to date on cutting edge developments of ML in geophysics.


**Summary And Contributions:**

Inference of underground structure from seismic measurements is a challenging problem with many important applications. Recently, machine learning techniques have been developed for this inverse problem, and this paper contributes to that body of work.

The authors present a large dataset containing many simulated seismic recordings from many simulated environments, grouped into 4 classes (or 11 sub-classes, one of which contains a 3D environment and a 2D grid of simulated sensors).

They also state an intention to release the code for 4 deep learning tools to infer the environment structure from the recorded data (at time of writing the github link in the paper is broken).  For now, they show performance metrics from the 4 tools on each dataset sub-class.

Finally, the authors rank the "complexity" of the datasets by 3 metrics and thereby confirm that the largest inversion errors they report are due to more complex environments. And the authors test "generalization" by training one of their inverse methods on one of the 10 "2D" subclasses of data, and testing performance on the other 9.

All in all, this work would be a strong contribution and aid to the field, if the code is indeed released.  My strongest criticisms are of the writing, which I believe makes this submission much harder to understand and interpret than needed.

---

> ### Author Response · Authors · 2022-08-27
> **Response to reviewer 7wdh [Part 2]**
>
> **Q7.1: Which training set should one use to train an inverse model to work on real-world data? Would our decisions change if we had some prior knowledge on the likely presence/absence of fault-lines, or a crude estimate of sub-surface "complexity"? If we used multiple inverse models trained on different data synthetic sets and found different sound-speed maps how might we decide which to trust?**
>
> The selection of the best inversion model is to minimize the discrepancy between the predicted velocity map and ground truth, which can be written as: $\underset{m}{\arg\min} D(c_{pred},c_{true})$, where $D$ is the discrepancy, $c_{pred}$ is the predicted velocity map given by inversion model ${m}$ and $c_{true}$ is the ground truth. The discrepancy $D$ may be obtained by calculation ($\ell_1$ or $\ell_2$ norm) or may be obtained from domain experts.
>
> In the real-world case, $c_{true}$ may not be available . If we have a prior velocity map $c_{prior}$, we will be able to calculate $\underset{m}{\arg\min} D(c_{pred},c_{prior})$ to approximate the equation above. However, the inversion models must be trained to give $c_{pred}$. In order to save computational cost, we can minimize the discrepancy between the training set and the prior velocity map:
> $ \underset{m}{\arg\min}\sum_{i} D(c_{i},c_{prior})$, where $i$ is the index of the training samples corresponding to $m$.
>
> If there is no prior information, we would suggest using the seismic loss as the discrepancy, which is similar to physics-driven FWI: $\underset{m}{\arg\min} ||f(c_{pred}),d_{true}||^2$,
> where $f$ is the forward modeling operator and $d_{true}$ is the observed seismic data. Below is the demonstration of this scheme.
>
>   - **Demonstration of choosing dataset for target in the real scenario:** Firstly, we choose a real velocity map in the Gulf of Mexico and generate its seismic data. Then we apply all twenty models trained across ten datasets (two models trained per dataset using $\ell_1$ and $\ell_2$ norm, respectively). We choose the two models with minimal loss.
>
>   - **Conclusion:** The best model trained using $\ell_1$ is from the FlatVel-B dataset and the best model trained using $\ell_2$ is from the Style-A dataset. Both models generate reasonably good velocity maps.
>
>   - **Details and Illustration:** Here we perform a test to simulate a realistic situation. We extract the inversion result of Golf of Mexico data [1] and simulate the seismic data as pseudo-real data. The pretrained VelocityGAN models with the OpenFWI dataset are applied to the pseudo-real data. The RMSE between the predicted seismic data and the pseudo-real data are given at https://tinyurl.com/2pav9kvp. The predicted velocity maps given by the models with the lowest  RMSE  look promising.
>
> [1] Huang, Y. and Schuster, G.T., 2018. Full‐waveform inversion with multisource frequency selection of marine streamer data. Geophysical Prospecting, 66(7), pp.1243-1257.
>
> All of the methods mentioned above would require some additional effort to select the best datasets to use. If we would like to choose one with minimal effort, we would suggest using the "Style Family" in that this particular dataset yields highly diversified features obtained from natural images. That would enable the inversion of field data in general cases. You may choose version A or version B depending on the requirement of the velocity resolution.
>
>
> **Q8: If possible, adding a metric of computational cost to Table 2 would help a reader recognize and understand the trade-off between performance and cost.**
>
> The computational cost metric is added in the main body as Table 5.
>
> **Q9: The link in the paper for the code is broken. Also the "Resources -> tutorial" link seems broken on the webpage.**
>
> Thank you for pointing this out. We have fixed all the broken links. The following are resources that would facilitate reproducibility.
>
> - Datasets URL: https://openfwi-lanl.GitHub.io/docs/data.html#vel
> - GitHub Repository: https://GitHub.com/lanl/openfwi
> - Pre-trained models: https://tinyurl.com/bddzkxfz
> - Tutorial: https://openfwi-lanl.GitHub.io/tutorial/
> - Google Group: https://groups.google.com/g/openfwi

---

> ### Author Response · Authors · 2022-08-27
> **Response to reviewer 7wdh [Part 1]**
>
> Thank you for your suggestions, we admit that the previous version of the paper is not friendly to readers without any geophysical background. To increase its readability, we have made several modifications following your suggestions below:
>
> **Q1: The figures (at least the first appearance of each kind) need axes-labels.**
>
> We have added axis labels to Figure 1 (see link: https://tinyurl.com/mtu8nsu4 )
>
>
> **Q2.1: Given that $p$ does not appear frequently in the text, just leave the explicit dependence.**
>
> We have rewritten equation 1. Only simple $p$ is introduced in the equation, its spatial and temporal dependencies are described in the text. Please refer to the paragraph in this link: https://tinyurl.com/yj6pttks
>
>
> **Q2.2: Equation 2 has dropped the dependence on the source properties $s(r, t)$. This should be stated explicitly**
>
> We have emphasized that this study focuses on the controlled source methods and the source function is given. Please refer to the paragraph in this link: https://tinyurl.com/yj6pttks
>
>
> **Q2.3: In the future work section, the authors might acknowledge that there is a subfield of "passive geophysical inference" that makes use of more complex, and potentially unknown sources.**
>
> - **Experiment:** We take P-wave arrival picking as an example of passive seismic problems. We convert the Style-B dataset by adding (a) the labels of P-wave arrivals, and (b) Gaussian noise to the seismic traces. Then we train an InversionNet to predict the P-wave arrivals from noisy seismic traces.
>
>   - **Conclusion:** The InversionNet trained on the converted dataset accurately recognizes P-wave arrivals.
>
>   - **Details:** We add Gaussian noise ($\sigma=5\times10^{-3}$) to the seismic data in Style-B dataset and train the InversionNet. The results of P-wave arrival picking are shown at https://tinyurl.com/bde8e6eb. We generate P-wave arrival labels by finding the first time step of an amplitude larger than 1. The experiment is added to the supplementary section 15, and the potential contributions of our datasets to the passive seismology community are summarized at https://tinyurl.com/yc52npj7.
>
> **Q3: I would strongly recommend starting with the current section 3.3 use-cases and then putting a column of "domain interests" or "applications" in table 1.**
>
> We have rewritten Section 3 and introduced the dataset according to the domain interests. Please refer to the paragraph in this link: https://tinyurl.com/y66y75r7. The "domain interest" column is also added to the table.
>
> **Q4: I would recommend moving section 3.2, on how the velocity maps were synthesized to the supplemental materials.**
>
> Section 3.2 is now moved to the Supplementary Material.
>
> **Q5: Fig. 4 is impressive, but readers would benefit from seeing what more dramatic failures look like.**
>
> Failure cases have been added to Fig. 4 (see link: https://tinyurl.com/5afx2p5s).
>
> **Q6: I would suggest three scatter plots of "complexity metric" vs. SSIM, with the different inverse models shown in different colors.**
>
> We add three scatter plots of each complexity metric VS SSIM covering three methods. The data samples are collected from 8 datasets in Vel and Fault family. With linear regression on each metric and the average SSIM, we obtain a straight line that decreases monotonically, which aligns with the claim that the performance drops as the complexity increases. We insert this plot into the main body and shift the table with concrete numerical results to the supplementary materials.
>
> The plots can be accessed through the link: https://tinyurl.com/xbe78ks7

---

> ### Author Response · Authors · 2022-08-28
> **Updates on Additional Experimental Results**
>
> Based on all reviewers' comments, we have completed **additional experiments on seven topics**, including physics-driven methods, robustness test, uncertainty quantification, more 3D experiments, etc. Each one is accompanied by quantitative results and illustrations. Please refer to [Summary of Response](https://openreview.net/forum?id=7w-a8PYPlP&noteId=ZmHUoyPU-mC) (part 1-4) for the descriptions, conclusions and details of the experiments. Feel free to let us know if you have additional questions.
>
> We will upload the final version of manuscript by the end of Aug 28.

---

> ### Author Response · Authors · 2022-08-29
> **Revision Submitted**
>
> We have uploaded the revised manuscript and the supplementary materials.
>
> The manuscript is improved according to reviewers' suggestions, including re-arrangement, more discussion and illustrations, and additional experiments. We have summarized the major updates in the [Summary of response](https://openreview.net/forum?id=7w-a8PYPlP&noteId=ZmHUoyPU-mC).
>
> Please feel free with any questions.

---

### Official Review · Reviewer_3jMn · 2022-07-28
**One of a kind large-scale dataset for full waveform inversion**

**Rating:** 7
**Confidence:** 3
**Correctness:** Dataset generation concept seems corr…
**Clarity:** Paper is well written and well organi…

**Strengths:**

1) Large-scale dataset solving the problem of lack of large-scale public dataset in FWI.
2) Contributions and explanation of concepts are solid.
3) Experiments are clear and extensive.


**Weaknesses:**

More experiments from 3D dataset expected.


Authors should include a thorough comparison with pervious FWI datasets, highlighting their contributions and how their dataset is unique from others (preferably in a tabular format). Ignore this comment if that is beyond the scope of this paper.

**Additional Feedback:**

The authors have done a good work presenting a large-scale dataset in FWI.

**Documentation:**

URL provided and data will be released soon as seen from author's comments.

**Ethics:**

No negative social impact.

**Relation To Prior Work:**

Related to prior work described adequately with proper citations.

**Summary And Contributions:**

The authors develop an open-source platform containing twelve datasets for full waveform inversion (FWI) and experiment on 4 deep learning methods, 3 supervised and 1 unsupervised. Through this benchmark, the authors try to solve the challenge of lack of large public datasets in the FWI community. The dataset considers diverse domains in geophysics , covers different geological sub-surface structures and contains huge amount of data ranging from 2K to 67K. It also has a dataset for 3D FWI. The paper also provides a generalization study to show the generalization capacity of deep learning methods across different domains.

---

> ### Author Response · Authors · 2022-08-25
> **Response to reviewer 3jMn**
>
> Thank you for the constructive comments.
>
> **Q1: More experiments from 3D dataset expected.**
>
> - **Experiment 1 (Benchmark based on $\ell_2$ loss):** We train the InversionNet3D with $\ell_2$ loss and compare it with the performance by $\ell_1$ loss, which has been benchmarked in the original submission.
>
>   - **Conclusion:** Compared to $\ell_1$ loss, $\ell_2$ loss leads to a degradation on SSIM of 3% (0.9837 vs 0.9462)
>
>   - **Detailed results:** The comparison of performance is provided at https://tinyurl.com/yt4f667n, while the qualitative illustration is shown in https://tinyurl.com/y57arbsr.
>
> - **Experiment 2 (Physics-driven VS Data-driven in 3D scenarios):** We compare physics-driven methods with data-driven methods in 3D scenarios
>
>   - **Conclusion 1:** Data-driven methods have better performance than physics-driven methods.  The SSIM of InversionNet3D on the FlatVel-A dataset is 68% better than the result by physics-driven methods (0.9837 vs 0.5861).
>
>   - **Conclusion 2:** Data-driven methods have computational advantages when the ratio between the number of train samples and test samples is less than 14,500. Typically, the ratio is 10 in our 3D Kimberlina-V1 dataset and data-driven methods are 1,450 times faster than the physics-driven methods.
>
>   - **Details and illustrations of Experiment 2**: The quantitative results and illustrations are given at https://tinyurl.com/2hndtf97 and https://tinyurl.com/2p8wxavz. Physics-driven methods are directly implemented on the test data while data-driven methods are trained on training data in the first place. The computation comparison between physics-driven and data-driven methods is related to the ratio between the number of train samples and test samples. The comparison of the per-sample computational cost is shown in the table at https://tinyurl.com/yksamx2u and we discuss the relationship between the total data-driven/physics-driven computation cost and train/test ratio in https://tinyurl.com/2sckv9t3.
>
> - **Experiment 3 (3D Noise Test):** We test the robustness to noise of Inversionnet in 3D scenarios
>
>   - **Conclusion 3:** InversionNet3D is robust to noise compared to InversionNet in 2D. The SSIM of the noise tests using InversionNet3D degrades by a maximum of 1.46%, which is much smaller than that of the InversionNet (up to 21.99%).
>
>   - **Detailed statistics of Experiment 3:** The detailed statistics can be found at https://tinyurl.com/4mebaadw
>
> - **Experiment 4 (3D Simulation-2D Slices Test):** We train an InversionNet with 2D velocity/seismic data slices of the 3D Kimberlina-V1 dataset (3D simulation, 2D slices) and compare the performance with the InversoinNet3D benchmark.
>
>   - **Conclusion:** InversionNet performance is comparable with the InversionNet3D benchmark on the 3D Kimberlina-V1 dataset. The SSIM of InversionNet (3D simulation, 2D slices) is up to 0.9652 compared to 0.9838 of InversoinNet3D.
>
>   - **Detailed statistics:** The detailed statistics can be found at https://tinyurl.com/crnpw6k5
>
> - **Experiment 5 (2D Simulation-2D Slices Test):** We cut 2D velocity slices of the 3D Kimberlina-V1 dataset and generate corresponding 2D seismic data with the benchmark forward modeling operator (2D simulation). Then we train an InversionNet with the generated samples and compare the performance with the InversoinNet3D benchmark.
>   - **Conclusion:** InversionNet performance is slightly better than the InversionNet3D benchmark on the 3D Kimberlina-V1 dataset. The SSIM of InversionNet (2D simulation) is up to 0.9974 compared to 0.9838 of InversoinNet3D.
>
> **Q2: Authors should include a thorough comparison with previous FWI datasets, highlighting their contributions and how their dataset is unique from others (preferably in a tabular format).**
>
> Previous datasets are not open, covering either 2D or 3D, capturing fewer geological structures on a single scale. In contrast, our *OpenFWI* datasets are *open-source*, covering *both 2D and 3D* scenarios, capturing *more* geological structures on *multiple* scales.
>
> The details of the comparison are provided via the following link: https://tinyurl.com/2p98vh35.
>
> **Q3: URL provided and data will be released soon as seen from author's comments.**
>
> Our datasets, codes, pre-trained models, and a tutorial are all publicly available now. We also have a Google Group to discuss issues about *OpenFWI*. We are open to designing more materials based on feedback from our users.
>
> - Datasets URL: https://openfwi-lanl.GitHub.io/docs/data.html#vel
> - GitHub Repository: https://GitHub.com/lanl/openfwi
> - Pre-trained models: https://tinyurl.com/bddzkxfz
> - Tutorial: https://openfwi-lanl.GitHub.io/tutorial/
> - Google Group: https://groups.google.com/g/openfwi

---

> > ### Comment · Reviewer_3jMn · 2022-09-01
> > **Response of Reviewer**
> >
> > My concerns for additional experiments in 3D dataset is addressed in details by the authors and for that I highly appreciate their efforts.
> > My decision for the paper is Accept.
> >
> > Thank You.

---

> ### Author Response · Authors · 2022-08-28
> **Updates on Additional Experimental Results**
>
> Based on all reviewers' comments, we have completed **additional experiments on seven topics**, including physics-driven methods, robustness test, uncertainty quantification, more 3D experiments, etc. Each one is accompanied by quantitative results and illustrations. Please refer to [Summary of Response](https://openreview.net/forum?id=7w-a8PYPlP&noteId=ZmHUoyPU-mC) (part 1-4) for the descriptions, conclusions and details of the experiments. Feel free to let us know if you have additional questions.
>
> We will upload the final version of manuscript by the end of Aug 28.

---

> ### Author Response · Authors · 2022-08-29
> **Revision Submitted**
>
> We have uploaded the revised manuscript and the supplementary materials.
>
> The manuscript is improved according to reviewers' suggestions, including re-arrangement, more discussion and illustrations, and additional experiments. We have summarized the major updates in the [Summary of response](https://openreview.net/forum?id=7w-a8PYPlP&noteId=ZmHUoyPU-mC).
>
> Please feel free with any questions.

---

### Official Review · Reviewer_d9sX · 2022-07-28
**Solid and valuable datasets and benchmarks for Full Waveform Inversion**

**Rating:** 8
**Confidence:** 2
**Correctness:** The methodology appears sound and cor…

**Strengths:**

•	There is a diverse set of data. Easy and hard versions are provided.

•	Benchmarks apply four deep learning techniques to the dataset.

•	The authors address generalizability using data based on style transfer. Results include a generalization study by training one dataset and testing on a different dataset.

•	Good explanations and good visuals provide intuition as to the content of the datasets.

•	Results include an evaluation of the complexity of the datasets.


**Weaknesses:**

There was a minor type, but this has been corrected.

I gave a Confidence rating of 2, since there may be some FWI domain-specific weaknesses that I am missing due to lack of experience in this area.

**Additional Feedback:**

N/A

**Clarity:**

Very well written. Clean organization. Even though I am not well-versed in this field, I found the paper easy to follow.

**Documentation:**

The code and datasets have been released, and they seem well-done and will enable reproducibility. I feel the benchmarks are documented to a sufficient degree of detail. The supplemental is well-written, and together with the main paper, makes for a comprehensive work.

**Ethics:**

No ethical concerns

**Relation To Prior Work:**

•	Good description of prior work.

•	Well established how this work differs from previous works.

•	Section 3.3 ties the datasets to several topics of interest.

**Summary And Contributions:**

The authors provide four families of datasets. There is diversity in the dataset sources. Two of the families are based on “math-driven” synthetic data, while a third dataset is based on simulations from a real reservoir. The final family uses an interesting technique to encourage diversity and generalizability: a style network transfers geologic styles onto “natural” images from the COCO dataset. The generalizability and complexity analysis provided by the authors helps enhance the value of the datasets.

---

> ### Author Response · Authors · 2022-08-24
> **Response to reviewer d9sX**
>
> **Q1: Typo in Line 38: “The existed seismic datasets…”**
>
> Thank you for pointing it out! The typo is corrected in the revised manuscript.
>
> **Q2: The code is not yet released, but given the quality of the paper, I will assume that it will enable reproducibility.**
>
> The codes are now available on GitHub. We also provide abundant resources to support reproducibility. Here are all the available resources:
>
> - Datasets URL: https://openfwi-lanl.GitHub.io/docs/data.html#vel
> - GitHub Repository: https://GitHub.com/lanl/openfwi
> - Pretrained models: https://tinyurl.com/bddzkxfz
> - Tutorial: https://openfwi-lanl.GitHub.io/tutorial/
> - Google Group: https://groups.google.com/g/openfwi
>
> In the meantime, to strengthen the manuscript, we add many additional experiments. We will update the summary of those experiments very soon. We are more than delighted to provide more insights if you have any concerns about the new results.

---

> > ### Comment · Reviewer_d9sX · 2022-08-25
> > **Thank you for the response**
> >
> > I have not had a chance to look through everything, but the code and documentation looks well-done. I will update my review.
> >
> > I did happen to see a typo in the README of https://github.com/lanl/openfwi:  convenient should be convenient.

---

> > > ### Author Response · Authors · 2022-08-28
> > > **Updates on Additional Experimental Results**
> > >
> > > Based on all reviewers' comments, we have completed **additional experiments on seven topics**, including physics-driven methods, robustness test, uncertainty quantification, more 3D experiments, etc. Each one is accompanied by quantitative results and illustrations. Please refer to [Summary of Response](https://openreview.net/forum?id=7w-a8PYPlP&noteId=ZmHUoyPU-mC) (part 1-4) for the descriptions, conclusions and details of the experiments. Feel free to let us know if you have additional questions.
> > >
> > > Also, we have fixed the typo, thanks for pointing it out.
> > >
> > > We will upload the final version of manuscript by the end of Aug 28.

---

> > > ### Author Response · Authors · 2022-08-29
> > > **Revision Submitted**
> > >
> > > We have uploaded the revised manuscript and the supplementary materials.
> > >
> > > The manuscript is improved according to reviewers' suggestions, including re-arrangement, more discussion and illustrations, and additional experiments. We have summarized the major updates in the [Summary of response](https://openreview.net/forum?id=7w-a8PYPlP&noteId=ZmHUoyPU-mC).
> > >
> > > Please feel free with any questions.

---

### Comment · Reviewer_GUDA · 2022-07-26
**Code release ?**

Will the benchmark code be available via GitHub ? This link from the dataset website is not valid - https://github.com/margarita-aicyd/OpenFWI.
Thank you.

---

> ### Author Response · Authors · 2022-07-26
> **Code release on Github**
>
> Thanks for raising the concern. We are waiting for approval from DOE / Los Alamos National Lab on releasing the codes and the pre-trained models. The Github repo link will be available once approved. Sorry for the confusion!

---

> ### Author Response · Authors · 2022-07-27
> **Code Release Schedule**
>
> Based on the discussion with the Lab, we plan to publish the pipeline codes including: (1) Data pre-processing, (2) Data loader, (3) Training routine, (4) Evaluation and other peripheral codes in early August. We will ensure the pipeline runs smoothly with training and testing the published datasets with a toy example model. According to the lab's reviewing schedule, we should be able to release the model codes of InversionNet and VelocityGAN by the end of August.
>
> Thank you!

---

### Author Response · Authors · 2022-08-24
**Summary of Response [Part 4]**

#### **e. Additional experiments on uncertainty quantification** (Reviewer XZjm)
- **Experiment 1: Quantifying uncertainty.** Due to the limited rebuttal duration, we follow [1] and conduct experiments on CurveVel-A to quantify the uncertainty of InversionNet as a case study.

  - **Conclusion:** The uncertainty on boundaries is higher than in other regions, which implies the prediction sensitivity around the boundaries. To quantify the correlation between the uncertainty and boundaries, we calculate the Pearson correlation coefficient between the uncertainty value and the gradient magnitude on the edge. The value is 0.5462, indicating a moderate positive correlation.

  - **Details and illustrations:** Following [1], we modify the network architecture by adding an additional dropout layer with a dropout rate of $p=0.2$ after each convolutional layer except the last one. The visualization of uncertainty is provided in the figure at https://tinyurl.com/ycx4bjbf. The relationship between uncertainty and the gradient magnitude on the edge is shown in the scatter plot (see link: https://tinyurl.com/5459u2v).
[1] Kendall, A., & Gal, Y. (2017). What uncertainties do we need in bayesian deep learning for computer vision?. Advances in neural information processing systems, 30.

- **Experiment 2: Quantifying uncertainty on noisy seismic data.** We compare uncertainty on noisy seismic data over multiple noise levels.
  - **Conclusion:** The uncertainty increases gradually when increasing the noise level.

  -  **Details and illustrations:** We add four levels of Gaussian noise with mean $\mu=0$ and different standard deviation $\sigma_{test}$ ranging from $1\times10^{-5}$ to $5\times10^{-4}$ to normalized seismic data in the test set. Quantitative results are shown in the table at https://tinyurl.com/5ysb8enj.

- **Experiment 3: Quantifying uncertainty for generalization test (cross datasets).** We train InversionNet on CurveVel-A and test it on all the other 2D datasets, excluding Kimberlina-CO$_2$ due to its different data dimensions.

  - **Conclusion:** The uncertainty values of cross datasets are much higher than training and testing on the same dataset, which indicates domain shifts lead to an increase in uncertainty.

  - **Details and illustrations:** Quantitative results are shown in the table at https://tinyurl.com/yc76dzfj.

#### **f. Code sharing and more details on the forward modeling** (Reviewer XZjm)

To further support data generation, we provide codes and details on the forward modeling, which is a necessary module (see link: https://tinyurl.com/3und483t and https://tinyurl.com/4xvejn2f). We develop our method based on this work: https://csim.kaust.edu.sa/files/SeismicInversion/Chapter.FD/lab.FD2.8/lab.html.

#### **g. Additional experiments on passive seismic** (Reviewer 7wdh)

- **Experiment:** We take P-wave arrival picking as an example of passive seismic problems. We convert the Style-B dataset by adding (a) the labels of P-wave arrivals, and (b) Gaussian noise to the seismic traces. Then we train an InversionNet to predict the P-wave arrivals from noisy seismic traces.

  - **Conclusion:** The InversionNet trained on the converted dataset accurately recognizes P-wave arrivals.

  - **Details:** We add Gaussian noise ($\sigma=5\times10^{-3}$) to the seismic data in Style-B dataset and train the InversionNet. The results of P-wave arrival picking are shown at https://tinyurl.com/bde8e6eb. We generate P-wave arrival labels by finding the first time step of an amplitude larger than 1. The experiment is added to the supplementary section 15, and the potential contributions of our datasets to the passive seismology community are summarized at https://tinyurl.com/yc52npj7.

#### **h. Demonstration of choosing a dataset for the target in the real scenario:** (Reviewer 7wdh)

- **Experiment:** We design a strategy to select the proper inverse model in real-world cases and validate it with pseudo-real data. Firstly, we choose a real velocity map in the Gulf of Mexico [1] and generate its seismic data. Then we apply all twenty models trained across ten datasets (two models trained per dataset using $\ell_1$ and $\ell_2$ norm, respectively). We choose the two models with minimal seismic loss.

  - **Conclusion:** The best model trained using $\ell_1$ is from the FlatVel-B dataset and the best model trained using $\ell_2$ is from the Style-A dataset. Both models generate reasonably good velocity maps.

  - **Details and Illustration:** The RMSE between the predicted seismic data and the pseudo-real data is given at https://tinyurl.com/5fs553uy. The predicted velocity maps and corresponding seismic data are given at https://tinyurl.com/353sa62e.

[1] Huang, Y. and Schuster, G.T., 2018. Full‐waveform inversion with multisource frequency selection of marine streamer data. Geophysical Prospecting, 66(7), pp.1243-1257.

Thank you all!

Best,
OpenFWI authors

---

### Author Response · Authors · 2022-08-28
**Summary of Response [Part 3]**

#### **c. Comparison of data-driven methods and physics-driven methods** (Reviewer XZjm)

- **Experiment 1 (Prediction accuracy):** We compare data-driven methods and physics-driven methods in terms of *prediction accuracy*

  - **Conclusion:** Data-driven methods have better performance than physics-driven methods. For example, the SSIM of data-driven methods on the CurveVel-A dataset is 24% better than the result by physics-driven methods (SSIM 0.8442 vs. 0.6816).

  - **Details and illustrations:** We perform the physics-driven methods on 500 samples from each dataset. The comparison between quantitative results of data-driven methods and physics-driven methods is given in https://tinyurl.com/4ph3by6h. The illustrations of the data-driven and physics-driven inversion results in each family are provided: [Vel Family](https://tinyurl.com/pjue2z9z), [Fault Family](https://tinyurl.com/294sps7e), [Style Family](https://tinyurl.com/3mubc7hh) and [3D Kimberlina-V1](https://tinyurl.com/2p8wxavz), respectively.

- **Experiment 2 (Computational cost):** Physic-driven methods and data-driven methods measure computational cost differently. Physics-driven methods don’t need training data but are directly optimized on the testing data for multiple epochs to generate the inversion results, while data-driven methods are trained on training data for multiple epochs and then applied to the test data in one trial. Thus, the computational comparison varies over different ratio between the number of training and test samples.

  - **Conclusion:** All three data-driven methods are faster when the ratio between the number of training and test samples is less than 62. Typically, the ratios for all twelve OpenFWI datasets are significantly lower (about 10). Therefore, data-driven methods speed up 6 times than the physics-driven methods. Note: data-driven would benefit from a large ratio between the number of training and test samples.

  - **Details and illustrations:** The pre-sample computational time of physics-driven methods and the average training time of data-driven methods for each sample are given at https://tinyurl.com/yksamx2u.  We discuss the relationship between the total data-driven/physics-driven computation cost ratio and train/test ratio for different datasets in https://tinyurl.com/y84cbe7t.

#### **d. Additional experiments on model robustness** (Reviewer XZjm)

- **Experiment 1: Adding noise on 2D test datasets.** Models are trained on 2D clean datasets but tested on noisy seismic data over multiple noise levels.
  - **Conclusion:** All models are robust when noise level is low. When the noise level increases, more degradation is observed in all models. Among them, InversionNet is the most sensitive to noise.
  - **Details and illustrations:** We add four levels of Gaussian noise with mean $\mu=0$ and different standard deviation $\sigma_{test}$ ranging from $1\times10^{-5}$ to $5\times10^{-4}$ to normalized seismic data in test set. The performance of each model is summarized in the table at https://tinyurl.com/y82cp7y6  and the figure at https://tinyurl.com/yckpauk5.

- **Experiment 2: Adding noise on the 3D test dataset.** Models are trained on 3D clean datasets (without noise) but tested on noisy seismic data over multiple noise levels.

  - **Conclusion:** Both the $\ell_1$- and $\ell_2$-trained InversionNet3D models are relatively robust to Gaussian noise with negligible degradation.

  - **Details and illustrations:** We add four levels of Gaussian noise with mean $\mu=0$ and different standard deviation $\sigma_{test}$ ranging from $1\times10^{-5}$ to $5\times10^{-4}$ to normalized seismic data in test set. The performance of each model is also shown in the table at https://tinyurl.com/y82cp7y6.

- **Experiment 3:  Adding noise on both training and test 2D dataset.** Due to the limited rebuttal duration, we only train our models on CurveVel-A with noisy seismic data to verify if it can improve model robustness.

  - **Conclusion:** Introducing noise into training effectively improves model robustness for all three models when testing noise level is high. When testing noise level is lower than training noise level, results are mixed. Specifically, InversionNet trained with noisy data achieves better performance than the one trained on clean data, while the other two models (VelocityGAN and UPFWI) have slight performance drop.

  - **Details and illustrations:** We train our models on CurveVel-A with two levels of Gaussian noise ($\sigma_{train}=1\times10^{-4}$ and $\sigma_{train}=1\times10^{-4}$). The standard deviation of the noise added to the test set $\sigma_{test}$ranges from $1\times10^{-5}$ to $5\times10^{-4}$. Quantitative results are shown in the table at https://tinyurl.com/2p97ayvn . Visualization of results is provided at https://tinyurl.com/25rw68yp.

---

### Author Response · Authors · 2022-08-28
**Summary of Response [Part 2]**

### 2. Summary of major updates:

#### **Paper arrangement**

We have revised the manuscript according to the comments. We plan to rearrange the manuscript together with everything ready. The revised manuscript and supplementary materials will be uploaded by Aug 28.

---

#### **Additional Experiments and contents**

#### **a. Additional experiments on 3D dataset** (By Reviewer 3jMn)
We provide five additional 3D experiments, which are illustrated below:

- **Experiment 1 (Benchmark based on $\ell_2$ loss):** We train the InversionNet3D with $\ell_2$ loss and compare it with the performance by $\ell_1$ loss, which has been benchmarked in the original submission.

   - **Conclusion:** Compared to $\ell_1$ loss, $\ell_2$ loss leads to a degradation on SSIM of 3% (0.9837 vs 0.9462)

   - **Detailed results:** The comparison of performance is provided at https://tinyurl.com/yt4f667n, while the qualitative illustration is shown at https://tinyurl.com/y57arbsr.

- **Experiment 2 (Physics-driven VS Data-driven in 3D scenarios):** We compare physics-driven methods with data-driven methods in 3D scenarios

   - **Conclusion-1:** Data-driven methods have better performance than physics-driven methods.  The SSIM of InversionNet3D on the FlatVel-A dataset is 68% better than the result by physics-driven methods (SSIM 0.9837 vs 0.5861).

  - **Conclusion-2:** Data-driven methods have computational advantages when the ratio between the number of train samples and test samples is less than 14,500. Typically, the ratio is 10 in our 3D Kimberlina-V1 dataset and data-driven methods are 1,450 times faster than the physics-driven methods.

  - **Details and illustrations**: The quantitative results and illustrations are given at https://tinyurl.com/2hndtf97 and https://tinyurl.com/2p8wxavz. Physics-driven methods are directly implemented on the test data while data-driven methods are trained on training data in the first place. The computation comparison between physics-driven and data-driven methods is related to the ratio between the number of train samples and test samples. The comparison of the per-sample computational cost is shown in the table at https://tinyurl.com/yksamx2u and we discuss the relationship between the total data-driven/physics-driven computation cost ratio and train/test ratio in https://tinyurl.com/2sckv9t3.

- **Experiment 3 (3D Noise Test):** We test the robustness to noise of Inversionnet in 3D scenarios

  - **Conclusion:** InversionNet3D is robust to noise compared to InversionNet in 2D. The SSIM of the noise tests using InversionNet3D degrades by a maximum of 1.46%, which is much smaller than that of the InversionNet (up to 21.99%).

  - **Detailed statistics:** The detailed statistics can be found at https://tinyurl.com/4mebaadw

- **Experiment 4 (3D Simulation-2D Slices Test):** We train an InversionNet with 2D velocity/seismic data slices of the 3D Kimberlina-V1 dataset (3D simulation, 2D slices) and compare the performance with the InversoinNet3D benchmark.

  - **Conclusion:** InversionNet performance is comparable with the InversionNet3D benchmark on the 3D Kimberlina-V1 dataset. The SSIM of InversionNet (3D simulation, 2D slices) is up to 0.9652 compared to 0.9838 of InversoinNet3D.

  - **Detailed statistics:** The detailed statistics can be found at https://tinyurl.com/crnpw6k5

- **Experiment 5 (2D Simulation-2D Slices Test):** We cut 2D velocity slices of the 3D Kimberlina-V1 dataset and generate corresponding 2D seismic data with the benchmark forward modeling operator (2D simulation). Then we train an InversionNet with the generated samples and compare the performance with the InversoinNet3D benchmark.
  - **Conclusion:** InversionNet performance is slightly better than the InversionNet3D benchmark on the 3D Kimberlina-V1 dataset. The SSIM of InversionNet (2D simulation) is up to 0.9974 compared to 0.9838 of InversoinNet3D.

  - **Detailed statistics:** The detailed statistics can be found at https://tinyurl.com/crnpw6k5

#### **b. Comparison of *OpenFWI* datasets and previous datasets** (By Reviewer 3jMn)

- Previous datasets are not open, covering either 2D or 3D, capturing fewer geological structures on a single scale. In contrast, our *OpenFWI* datasets are *open-source*, covering *both 2D and 3D* scenarios, capturing *more* geological structures on *multiple* scales.

- The details of the comparison are provided via the following link: https://tinyurl.com/2p98vh35.

---

### Author Response · Authors · 2022-08-28
**Summary of Response [Part 1]**

Dear reviewers,

We are grateful for the valuable comments and fruitful thoughts. *OpenFWI* is expected to inspire more work to facilitate diverse research efforts on __AI for science__ initiatives.

### Broader Impact

**Data-driven FWI:** FWI is a typical scientific problem being studied with physics-driven approaches for decades. With the rapid development of deep learning, we have seen a myriad of data-driven approaches. *OpenFWI* embraces this junction and brings the community with the potential of (1) *Unified Evaluation*,  (2) *Further Improvement* and (3) *Re-producibility and Integrity*, which are essential as the study on this topic evolves.

**Future Developments:** We plan to maintain *OpenFWI* meticulously by releasing new datasets, and new benchmarks and serving the community with follow-up questions. There will be workshops with future updates about *OpenFWI*, and data competitions with more challenging data/tasks at the appropriate junction.

**AI for science:** Scientific machine learning (SciML) is demonstrating its great potential in various disciplines including geoscience. Compared to other fields in Machine Learning, serious data challenges remain: sparse direct measurements; unbalanced data distribution; inevitable noise, etc. Our efforts would hopefully shed some light on how to overcome those challenges for SciML to enable exciting progress in typical science-rich and data-starved scientific fields.

Based on the raised concerns, we added many extra experiments, illustrations, and analyses, and most importantly, __the codes are now released.__  We respond to the concerns and comments of each reviewer individually. Here we summarize the major updates of the manuscript and the new features to our *OpenFWI* platform as follows:



---
### 1. Code, pre-trained models, and tutorials are released! (Concern by all reviewers)

The codes and pre-trained models of InversionNet and VelocityGAN are now public on GitHub. We add the OSS license and BSD-3 license, as required by the Los Alamos National Laboratory (LANL) and the U.S. Department of Energy (DOE), U.S.A. We also design a beginner-friendly tutorial and share it on our website. The following are all the publicly available resources that support the reproducibility of OpenFWI.

- Website: https://openfwi-lanl.GitHub.io
- Datasets URL: https://openfwi-lanl.GitHub.io/docs/data.html#vel
- GitHub Repository: https://GitHub.com/lanl/openfwi
- Pre-trained models: https://tinyurl.com/bddzkxfz
- Tutorial: https://openfwi-lanl.GitHub.io/tutorial/
- Google Group: https://groups.google.com/g/openfwi

---

### Meta-Review · Area_Chair_N66B · 2022-09-07

**Recommendation:** Accept
**Confidence:** 4

**Metareview:**

All reviewers agreed that the proposed benchmark datasets for multi-structural seismic full waveform inversion are solid and valuable, and  there is diversity in the dataset sources. The baselines and metrics are clear. It may well encourage the research in this direction, and also inspire future open-source efforts on AI for science.

It is a clear acceptance. Based on the overall comments from all reviewers, it is recommended for Spotlight.

---

### Decision · Program_Chairs · 2022-09-16

Accept